# Effect of Sintering Temperatures, Reinforcement Size on Mechanical Properties and Fortification Mechanisms on the Particle Size Distribution of B_4_C, SiC and ZrO_2_ in Titanium Metal Matrix Composites

**DOI:** 10.3390/ma15165525

**Published:** 2022-08-12

**Authors:** Birhane Assefa Gemeda, Devendra Kumar Sinha, Gyanendra Kumar Singh, Abdulaziz H. Alghtani, Vineet Tirth, Ali Algahtani, Getinet Asrat Mengesha, Gulam Mohammed Sayeed Ahmed, Nazia Hossain

**Affiliations:** 1Department of Mechanical Engineering, Program of Mechanical Design and Manufacturing Engineering, School of Mechanical, Chemical and Materials Engineering, Adama Science and Technology University, Adama 1888, Ethiopia or; 2Department of Mechanical Engineering, College of Engineering, Taif University, P.O. Box 11099, Taif 21944, Makkah, Saudi Arabia; 3Department of Mechanical Engineering, College of Engineering, King Khalid University, Abha 61421, Asir, Saudi Arabia; 4Research Center for Advanced Materials Science (RCAMS), King Khalid University, Guraiger, P.O.Box 9004, Abha 61413, Asir, Saudi Arabia; 5Department of Materials Science and Engineering, Adama Science and Technology University, Adama 1888, Ethiopia; 6Center of Excellence (COE) for Advanced Manufacturing Engineering, Program of Mechanical Design and Manufacturing Engineering, School of Mechanical, Chemical and Materials Engineering, Adama Science and Technology University, Adama 1888, Ethiopia; 7School of Engineering, RMIT University, Melbourne, VIC 3001, Australia

**Keywords:** titanium matrix composites, particulates reinforcements, B_4_C, SiC, ZrO_2_

## Abstract

Titanium metal matrix composites/TMMCs are reinforced ceramic reinforcements that have been developed and used in the automotive, biological, implants, and aerospace fields. At high temperatures, TMMCs can provide up to 50% weight reduction compared to monolithic super alloys while maintaining comparable quality or state of strength. The objective of this research was the analysis and evaluation of the effect/influence of different sintering temperatures, reinforcement size dependence of mechanical properties, and fortification mechanisms on the particle size distribution of B_4_C, SiC, and ZrO_2_ reinforced TMMCs that were produced and fabricated by powder metallurgy/PM. SEM, XRD, a Rockwell hardness tester, and the Archimedes principle were used in this analysis. The composites’ hardness, approximation, tensile, yielding, and ultimate strength were all increased. As the composite was reinforced with low-density ceramics material and particles, its density decreased. The volume and void content in all the synthesized specimens is below 1%; this is the result of good sample densification, mechanical properties and uniform distribution of the reinforced particle samples; 5% B_4_C, 12.5% SiC, 7.5% ZrO_2_, 75% Ti develop higher mechanical properties, such as higher hardness, approximation tensile, yielding, and ultimate strength and low porosity.

## 1. Introduction

Metal matrix composites (MMCs) are gaining popularity in scientific and industry circles due to their appealing physical, and mechanical qualities and have tremendous potential for use in the automobile and aerospace industries [1]. When compared to conventional and homogeneous metal alloys [2], particle-armored MMCs have superior mechanical properties, such as strengthening and stiffening [3,4], hardness [5,6], and fracture toughness [7,8]. As a result, composites have the ability to deliver customized mechanical characteristics, making them appealing for a broad array of applications [9]. To enhance the properties of particulate-armored MMCs, recent studies have focused on decreasing reinforcement particles from the micrometric domain towards the sub-nano and micro-metric size scales [6,10]. Several studies identified that decreasing the size/dimension of armored plate particles can increase the strength and minimize stress concentration at the corners of nanoparticle-reinforced composites, resulting in increased work hardenability due to the armored plate dislocation effect [11,12,13,14]. Many researchers are intrigued by their uses in aerospace, automotive, chemical, biomedical [15], and other industries. In the aforementioned applications, the demand for greater quality materials with improved mechanical, tribological [14] and machining [16] properties has increased dramatically. Pure titanium is now the most appealing metallic material for aerospace and vehicle applications due to its low density (4.6 g/cm^3^), excellent strength to weight ratios, and great corrosion resistance. Titanium has poor mechanical characteristics and it is brittle and easily fractures at room temperature. Pure elements lack the ability to withstand force or chemical attack and must be combined with additional components to balance the physical and mechanical qualities. When numerous components are combined, they form a material with unique properties for each individual component [17,18]. Powder metallurgy/PM is the most effective method for producing homogenous composite materials [19,20,21,22,23]. Titanium dioxide can be found in three polymorphs, including anatase, brookite, and rutile. The anatase phase is metastable and can be converted to rutile by heating treatment. Rutile mode thermodynamics had much greater stability than brookite and anatase at heating and depressed the room for macrocrystalline systems; under pellet conditions, the rutile structure is described to be thermodynamically stable. Several types of particles, whiskers, or fiber ceramics that can be used as reinforcement in composites have been proposed in previous works to improve the overall properties of Titanium matrix composites, such as Ti_5_Si_3_, SiC, TiO_2_, Al_2_O_3_, TiB, TiC, graphene nanoplatelet, nanodiamonds, WC, ZrO_2_, B_4_C, MoS_2_, rare earth oxides, such as La_2_O_3_, Y_2_O_3_, and Nd_2_O_3_, etc., refs. [24,25,26,27,28,29,30,31,32,33,34] according to the required application of the materials [22,35,36,37,38,39,40]. Among these candidates, B_4_C, SiC, and ZrO_2_ to titanium base metal have been considered the best reinforcements due to their good compatibility with matrix alloys, particularly the coefficient of friction, micro hardness, wear resistance, corrosion resistance, yield strength, tensile strength, ultimate tensile strength, tensile failure strength, and compressive failure strength, formability, toughness, and biocompatibility. Titanium matrix composites/TMCs/fortified with higher strength and stiffness, than ceramic particles/whiskers have emerged as one of the most promising materials in the automotive and aerospace industries due to excellent properties with higher specific strength and stiffness, damage tolerance, and preferable mechanical characteristics at elevated temperatures [38,39,40]. TMCs, in particular, have found widespread application in the aforementioned areas due to their high specific strength, specific stiffness, and outstanding mechanical characteristics under high heat and temperature [5]. Because of the excellent and remarkable features, such as good hardness, low density, high tensile and compressive strength, high toughness, and excellent machinability, the particulate reinforcements of B_4_C, SiC, and ZrO_2_ reinforced metal composites have been recognized as potential material requirements for such applications, indicating a potential use in a several ranges of high-stress applications. The primary rationalizations for using titanium in the aerospace sector are weight savings, particularly as a steel potential substitute; space constraints to substitute Al alloys; thermal resistance and efficiency to substitute Al, Ni, steel alloys; higher corrosion resistance to substitute Al and low alloy steels and composite compatibility to substitute Al alloys. As a result, titanium metal matrix is a popular aerospace engineering material. This material is a promising candidate and significant in addressing the scarcity and limitations of other super alloy monotonic materials. A study has suggested that lowering the dimension of the reinforcement detritus to 100 nm may improve the strength and ductility of synthesized and development MMCs. Despite the enhancement in nanoparticle MMCs, many major questions remain unresolved. To make use of the benefits of smaller reinforcing particles, such as reduced stress concentration, it is necessary to minimize/eliminate nanoparticle agglomerates and establish a homogeneous spatial contribution and spread of individual particles throughout the matrix. Second, it is acknowledged that the structure and chemistry of matrix/reinforcement interfaces have a significant impact on the mechanical properties, and cryo-milling reduces interfacial modifications by suppressing diffusion and chemical reactions at cryogenic temperatures and separating reactive nanoparticle powders from the environment. Third, B_4_C is intriguing because it has a very high hardness at room temperature, which is only slightly lower than that of cubic BN and diamond; at temperatures exceeding 1200 °C, its own hardness has been shown to exceed that of diamond. Furthermore, B_4_C is cheaper and less challenging to create than diamond and cubic BN. These properties, together with its high melting point, low density, and extraordinary chemical inertness, make B_4_C an excellent reinforcement for a wide range of metals and found that Ti composites containing nano dimension B_4_C particles had higher strength and better tensile ductility than those with micro dimensions B_4_C particles [41,42]. Fourth, zirconium dioxide (ZrO_2_) has outstanding biomechanical qualities, such as fracture strength, toughness, and fatigue resistance, low elasticity module and strength, as well as high wear resistance and bio-compatibility. Powder metallurgy has been stated as the method of combining, pressing, and sintering the ingredients of a composite. PM is the most effective production process and method for generating homogeneous composite materials. This approach produces exceptional characteristics by achieving good uniformity and low porosity. Fifth, SiC reinforcing was chosen as reinforcement due to its corrosion resistance, high strength, outstanding thermal stability, formability, ductility, stiffness, low cost, and other characteristics. In this study, powder metallurgy was utilized to synthesize Ti–B_4_C, SiC, and ZrO_2_ nanocomposites. The mechanical characteristics of the developed TMCs materials’ microstructure, densification, micro hardness, sintering temperature influence, and reinforcement size dependence and distribution have all been investigated.

## 2. Experimental Procedure and Work

Titanium with nanoparticles size 100 nm, purity > 90%, and reinforcement of nanoparticle powders of B_4_C, SiC, and ZrO_2_ with a particle size of 100 nm were acquired from METALFORT Company, Mumbai, India, and utilized as starting materials. To achieve this, powder metallurgy was used to make Ti-B_4_C, SiC, and ZrO_2_ nanocomposites in this study. These mechanical characteristics of the developed and produced TMCs materials, including microstructure, densification, micro hardness, sintering temperature influence, and reinforcement size dependence and distribution, have all been studied. The effect of reinforcement size and hardness was measured with a Precision weight balance and Rockwell hardness testing machine as shown in Figure 1; sintering temperature of B_4_C, SiC, and ZrO_2_ nanoparticles of 2.5%, 5%, 7.5%, and 12.5% reinforcement and base metal matrix titanium with different weight percentages of 75%, 77.5%, 82.5%, 85%, and 100% material were investigated. Sample notation and composition were tabulated in Table 1, powder metallurgy based approaches are appealing for the fabrication of both whisker- and particle-reinforced MMCs due to their simplicity in comparison to alternative manufacturing processes and the ability to generate complex structures with high precision [43,44,45]. The sintering and pressing method, also known as pressureless sintering, is the most fundamental and cost-effective traditional PM technology. The traditional PM processing approach is classified into three major steps: powder blending and mixing, cold compression, and sintering. Titanium and reinforcement of nanopowders of B_4_C, SiC, and ZrO_2_ powders were mixed with different weight percentages to create this power blend using high-speed dry ball milling and was employed; a 50 Mpa hydraulic press, shaped the powder into solid objects and sintered at a capacity of 1700 °C in a box furnace with different sintering temperatures. Table 2 represents the experimental procedure of Titanium MMM synthesized Sintered specimens having an average diameter of 20.0 mm and heights of 7 mm. The microstructure of the specimens was examined through scanning electron microscopy using the Jeol Japan SEM, Model JCM/6000PLUS BENCH TOP SEM, Musashino, Akishima, Tokyo 196-8558, JAPAN. A Shimadzu Corporation XRD-7000 Maxima X-ray diffractometer has also been used to analyze various phase compositions of the samples. A Digital Rockwell microhardness type HRS-150 was also utilized for microhardness measurement testing, Beijing United Tester Co., LTD of Beijing, China, and accomplished the examination with a weight of 150 kgf and a dwell time of 15 s. Figure 1 depicts the TMCs development and characterization flow chart.

### 2.1. Synthesis Titanium Metal Matrix Reinforced with Nano Particles by Powder Metallurgy Process

Powder metallurgy/PM/ is the best promising method for producing TMCs. Despite being a more expensive technique, it has the advantage of creating precision components without melting. The science of producing metal powders/particles and finished/semifinished items from mixed/alloyed powders with/without nonmetallic elements is known as powder metallurgy. Powder metallurgy consists of three distinct steps: (A) the combination of metal and reinforcing powders, (B) powder compaction/squeezing to form a green material body, and (C) sintering, which is frequently followed by additional processing.

#### 2.1.1. Mixing or Blending and Sizing of Powders

Blending is the procedure of combining/unifying powders/particles that have different particle/powder sizes and shapes by passing them through the same simple mechanism. Blending should be used to achieve a consistent distribution of particle sizes and reduce porosity [46,47]. High-speed dry ball milling was used to grind/blend Ti, SiC, ZrO_2_, and B_4_C nanoparticle powders. The ball mill was outfitted with a high-speed spindle and ran for two hours to produce a homogeneous powder mixture. The powders were then obtained with the desired grain size and appropriate for the subsequent process. The weight, with the combined total of the powders, is 5 gm. Because these Nano-particles were a uniform particle size of 100 m, a powder-to-ball weight of 1:5 is ideal for successful mixing and sizing for 2 h. The milled powder results of ten (10) samples of Ti-based reinforced composites were synthesized using a high-speed dry ball milling machine.

#### 2.1.2. Compaction

Utilizing a proper punch and die to generate green compacts using mechanical or hydraulic presses [46], powder mixtures are widely compacted. The powder combinations were cold crushing, squeezing, and compacting at an appropriate pressure with the use of a uniaxial press [48]. Compaction has been stated as the development of procedures, systems, and methods occurring in compacting and squeezing metallic particles in a hydraulically driven due to the required shape. These hydraulic presses have an owing capacity of 25 tons, a 200 mm diameter pressure plate, and a 150 mm ram stroke. To produce the specimen, milled metallic and ceramic powders are introduced into the die’s cavity. For 30 min, all prepared samples are compressed at 50 MPa. After being formed at room temperature, the product is a specimen discharged from the die cavity at room temperature. In this experiment, hydraulic press samples were used to condense the milling power. Following the completion with a hydraulic cylindrical compression lower type bucket elevator and a green compacted sample shape, the cylindrical shaped specimens were visible.

#### 2.1.3. Sintering

Sintering is defined as the procedure, method, and system for binding particles together by heating green compacts in a controlled environment. To sinter materials below their melting points, mesh belts, walking beams, pusher types, and batch furnaces are all used [46]. Many researchers assert that the highest sintering temperature is used to produce components with good surface finish and quality, and it has been demonstrated that as the sintering temperature rises, so do the material’s mechanical properties [49,50]. The titanium sintering process was kept at a temperature ranging from 750 to 1350 °C, according to [51,52]. Powder metallurgy operates in the solid state below the melting point in a material with a ratio greater than 0.5 times the melting temperature and less than 0.8 times the melting temperature. Sintering is the compacting and formation of sample solids through heating in a vacuum furnace. The compression rates for samples one through ten sintered in a vacuum box furnace for 2 h at room temperature were 900 °C, 950 °C, 1000 °C, 1050 °C, 1100 °C,1150 °C,1200 °C,1250 °C,1300 °C, and 1350 °C, respectively.

## 3. Results and Discussion

The effect of various sintering temperatures, reinforcement particle size distribution, the dependence of mechanical properties, and strengthening mechanisms in B_4_C, SiC, and ZrO_2_ reinforced titanium metal matrix composites of synthesized TMC material armored with B_4_C, SiC, and ZrO_2_ through powder metallurgy techniques and various characterization was experimentally examined. Manufactured TMCs specimens were developed according to the selected parameters. The experiment’s design was utilized to determine the optimal sintering temperature, compaction parameters, mixing parameters, and reinforcement particle size distribution that had the most influence on the mechanical and physical properties of the fabricated TMCs.

### 3.1. Characterization of the Synthesized TMCs

The surface topography of the synthesized specimen was investigated using SEM scanning to indicate the SEM microstructure of the synthesized TMMC along with base-Ti6Al4V specimens having a coarse lamellar + microstructure with phase separation created during sintering at high temperature and a subsequent slow cooling rate [5]. The Figure 2 shows the SEM micrographs of the samples.

It has been observed from SEM micrographs that (2.5% B4C, 7.5% SiC, 12.5% ZrO2, 77.5% Ti) by increasing the concentrations of ZrO_2_ and decreasing B_4_C and SiC particles in sample S3, the porosity decreased and the surface densified in the Ti-based metal matrix. In most places on the surface of the Ti-based metal matrix sample S3, the ZrO_2_ particle agglomeration is seen. Because increasing the ZrO_2_ particle caused agglomeration Muharrem Pul et al. [53]. The addition of ZrO_2_ particles in the metal matrix composite initiated agglomeration. The microstructure of S3 shows that there is no porous structure between the 77.5%Ti and 2.5% B_4_C; 7.5% SiC and 12.5% ZrO_2_ reinforcing particles and the bonding of the phases are very good. We observed that samples S6 and S7 have the same concentrations of ZrO_2_ as S3, however, due to the increasing concentrations of B_4_C and SiC, S6 and S7 are more porous microstructures. Harish et al. [54] report that the porosity of the metal matrix composite materials in the microstructure increases with the increase in the particles of reinforcing. Therefore, with the same concentration of ZrO_2_ and different concentrations of B_4_C and SiC, samples S3, S6, and S7 have different surface morphologies.

### 3.2. XRD Analysis

The elemental phases present in the manufactured samples were analyzed using XRD in accordance with the XRD working principle: Bragg’s law [55] the XRD was performed on a fully computerized powder X-ray diffractometer (XRD7000 X-ray DIFFRACTOMETER, SHIMADZU Corporation (Tokyo, Japan)) at 40KV and 30mA. The XRD spectrum was generated at a two-degree angle ranging from 10 to 85 degrees with a 0.02-degree step size and continuous scanning at three degrees per minute for 0.40 s. Miller indices (hkl) are used to identify various planes of atoms, and the observed diffraction peaks can be related to the planes of atoms to aid in atomic structure and microstructure analysis. When analyzing XRD data, we look for trends that correspond to crystal structure directionality by analyzing the Miller indices of diffraction peaks. The crystal structure determines the position and intensity of peaks in a diffraction pattern. The fabricated samples were subjected to XRD analysis to determine whether any intermetallic compounds were formed during the sintering process [56]. When the diffractometer is linked to the X’pert data collector software, d’ values are displayed directly on the diffraction pattern. These d’ values were then used to identify different phases using ASTM X-ray diffraction data cards. To confirm the presence of minor precipitate phases detected by the diffraction pattern, the d′ values for different phases were obtained using JCPDS cards included with the software and manually compared with the diffraction pattern of all samples [55].

Figure 3 represents the XRD graph of the titanium-based metal matrix composite powders with milling before compaction and sintering were performed. The graph shows that there is a dominance of the titanium matrix peaks, which ascribes that in the milling process there was an undesirable interfacial chemical reaction between the hybrid reinforcements and the matrix with less peak is detected with angles of 2θ = 27.60 and 54.480 corresponding to (110) and (211), respectively, and the rutile (TiO_2_) (JCPDScard number: 021-1276) developed. As shown in Figure 3a,b, titanium with a hexagonal closed packed crystal structure with a = b = c = 1.587 and ∝ = β = γ ≠ 90 with an experimental density of 4.6 g/cm^3^ can be detected in the titanium metal matrix samples, regardless of whether it is before or after sintering. However, in the composite shown in Figure 3b, the peaks corresponding to distinct phases are recognized as Ti, B_4_C, SiC, and ZrO_2_. The presence of Ti, B_4_C, SiC, ZrO_2_, and rutile (TiO_2_) in the titanium metal matrix is highly correlated with the presence of Ti, B_4_C, SiC, and ZrO_2_ in the titanium metal matrix. Variation of Rockwell micro hardness of fabricated specimen surface analysis shown in Figure 4. The peak of rutile (TiO_2_) (JCPDS card number: 021-1276) was shown for all samples at 2θ = 27.440, 41.60, 44.080, 79.90 and 84.340 corresponding to the (110), (002), (210), (212) and (400) crystallographic plane, respectively. The peak of SiC was shown at angles 2θ = 41.280, 62.780, and 76.60 corresponding to the (200), (110), and (103) crystallographic plane (JCPDS card number: 049-1623). The peaks of B_4_C at angles 2θ = 69.840 correspond to the (220) crystallographic planes. The peaks ZrO_2_ are also shown at angles of 2θ = 27.50 and 64.10, corresponding to (−111) and (211) crystallographic planes. This was observed after sintering the metal matrix of all components of the composite detected by X-ray. Synthesized TMC density in different composition of samples shown in Figure 5 and Figure 6 shows comparison of actual density and density predictable by ule of mixture.

### 3.3. Surface Hardness Testing of TMCs

Rockwell hardness examinations are the most extensively used hardness measuring techniques in the manufacturing sector. Diamond indenters are used to achieve various Rockwell hardness scales currently specified in ISO 6508-1, the most important of which are HRC, HRA, and HRN. The problems in introducing assessment methods to the measuring capabilities of the hardness test machines demonstrate the industry’s requirement for more accurate calibration techniques within Rockwell hardness investigation machines. The ASTM E18 and 28 standard testing procedures were used to determine the hardness number of the specimens using the Rockwell hardness tester scale as tabulated in Table 3. The indenter utilized was with 150 kg Brale. The load application time is 15 s [41]. Variation of Rockwell micro hardness is shown in Figure 4.

**Figure 4 materials-15-05525-f004:**
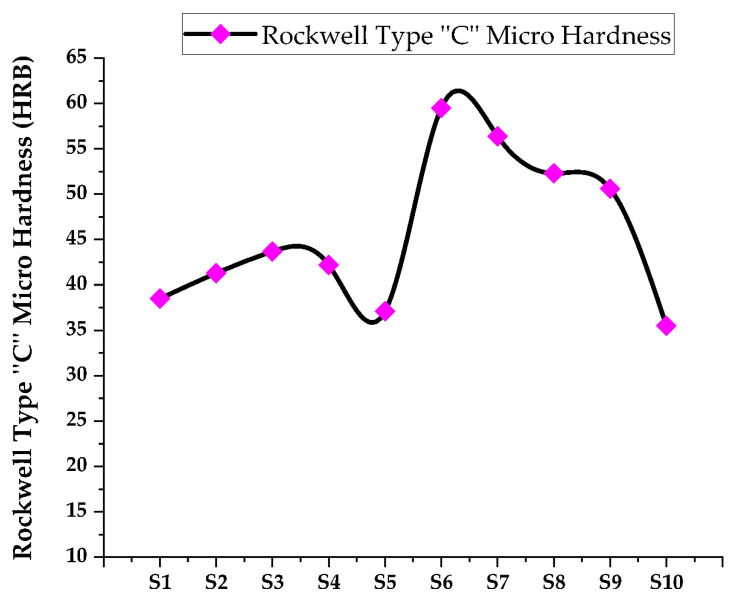
Variation of Rockwell micro hardness of fabricated specimen surface analysis.

According to Table 4, samples W% reinforcement and base metal by the law of mixture: the lower the hardness created in sintered specimens, the lower the minimum sintering temperature/heat. As a result, factors, such as insufficient reinforcement, particle dispersion, clustering of reinforced particles, temperature mismatch between particles and matrix, and particle size discrepancies between matrixes and reinforcing phases all affect the hardness of such composites. Hardness is caused by thermal mismatch, but clustering and insufficient dispersion can result in a decrease in hardness.

### 3.4. Density and Porosity Measurement

According to the density of the reinforcing material, the phase and size of the combining components, and the process of manufacturing the composite material, the density can increase or decrease [57]. Archimedes’ principle was used to estimate the bulk density, porosity, and water absorption of sintered samples. The specimen’s sintered weight was first determined using a precision digital weighing balance (HR-250AZ, A&D Company Limited, Korea). The drop in density can be attributed to the reinforcing particles’ decreased density and the creation of porosity [24]. The specimen was then immersed in 70 °C hot water for 2 h, and the soaked weight was calculated by ASTME Designations C20-00 and [24,37]. The extent to which the TMCs compacted and sintered was measured using a tumbler full of water into which the samples were suspended down inside the water.

Through the Archimedes principle, the weight of sintered, soaked, and submerged materials were examined following equation: [58]
(1)$$\mathrm{Bulk}\;\mathrm{density}=\frac{\mathrm{Sin}\mathrm{tered}\;\mathrm{weight}\;\left(\mathrm{gm}\right)}{\;\mathrm{soaking}\;\mathrm{weight}\left(\mathrm{gm}\right)\;–\mathrm{Suspension}\;\mathrm{weight}\;\left(\mathrm{gm}\right)}\times \rho w\;$$
(2)$$\mathrm{Archimedes}\;\mathrm{density}=\frac{\mathrm{Weight}\;\left(\mathrm{in}\;\mathrm{Air}\right)}{\;\mathrm{Weight}\;\left(\mathrm{in}\;\mathrm{Air}\right)–\mathrm{Weight}\;\left(\mathrm{in}\;\mathrm{water}\right)}\times \mathrm{Density}\;\mathrm{of}\;\mathrm{water}\left(\rho w\right)$$

To analyze the generated nanocomposites, the genuine density of all sintered specimens was tested using the Archimedes method and a density measurement device with a precision digital weighing balance (HR-250AZ, A&D Company limited, Seoul, Korea). The theoretical density was then computed using Agarwal and Broutman’s equation [59,60,61,62] given in Table 5,
(3)$$\rho \mathrm{th}=\frac{1}{\;\left(\frac{\mathrm{Wf}}{\rho f}\right)+\left(\frac{\mathrm{Wm}}{\rho m}\right)}\;$$
where Wf denotes the weight fraction of reinforcement, Wm is the weight fraction of Ti6Al4V, and denotes the theoretical density of the composite and represents the density of reinforcements in SiC (3.21 g/cm^3^), B_4_C (2.52 g/cm^3^), and ZrO_2_ (5.68 g/cm^3^); ρm represents the density of the Ti6Al4V matrix (4.43 g/cm^3^). The variation in the bulk density was then computed as illustrated in Figure 5. The high relative density of the sintered specimens implies that the constituent particles have strong interface bonding with negligible porosity or voids.

**Figure 5 materials-15-05525-f005:**
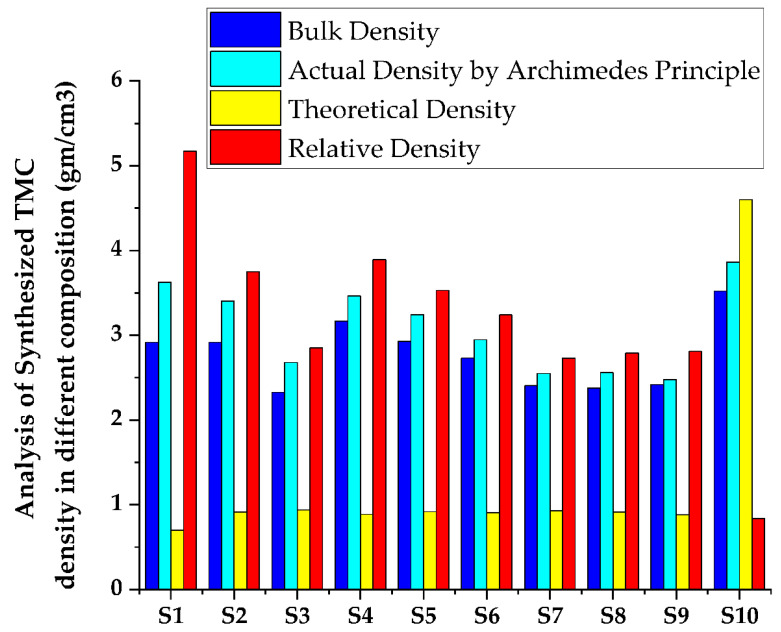
Synthesized TMC density analysis in different composition of samples.

**Figure 6 materials-15-05525-f006:**
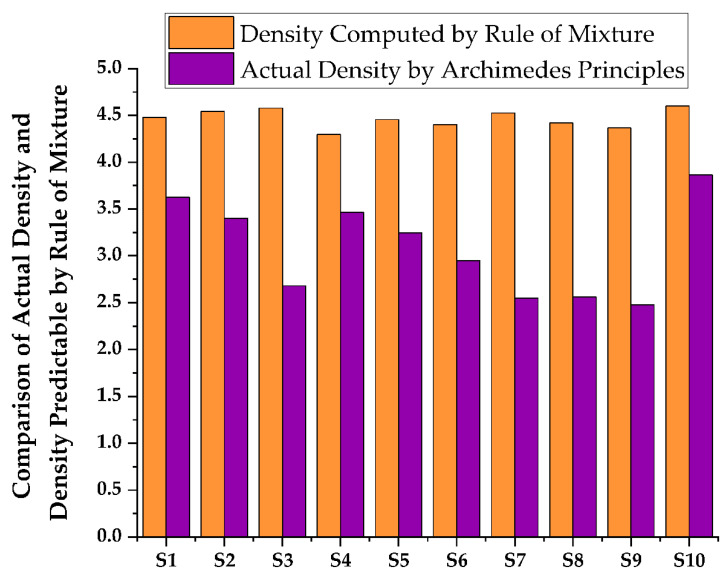
Comparison of actual density and density predictable by rule of mixture.

**Table 5 materials-15-05525-t005:** Different Sintered Samples Density Analysis.

No	Sintered Sample Composition	Bulk Density g/cm^3^	Actual Density by Archimedes Principles (gm/cm^3^)	Theoretical Density (gm/cm^3^)	Relative Density gm/cm^3^	Porosity Volume%	Porosity by Theoretical Density %	Porosity by Archimedes Principles %
1	S1	2.91734	3.62416	0.7	5.17	0.08277	8.27777	19.5029714
2	S2	2.91657	3.40144	0.9132	3.75	0.02551	2.55185	14.254711
3	S3	2.32648	2.68	0.9407	2.85	0.02212	2.21268	13.1906767
4	S4	3.16869	3.4635	0.888	3.89	0.03233	3.23365	8.51375733
5	S5	2.93055	3.24514	0.9174	3.53	0.02545	2.54533	9.69420259
6	S6	2.73120	2.94791	0.9086	3.24	0.03100	3.10050	7.35129068
7	S7	2.40370	2.54942	0.9311	2.73	0.02702	2.70256	5.71575695
8	S8	2.37668	2.55907	0.9132	2.79	0.03391	3.391854	7.12711242
9	S9	2.41874	2.47677	0.88	2.81	0.04845	4.84501	2.34297109
10	S10	3.52050	3.86419	4.6	0.84	0.16086	16.08695	8.61676746
Void content volume (%)								
0.8								
0.26								
0.22								
0.32								
0.25								
0.31								
0.7								
0.34								
0.48								
1.6								

Therefore, the Bulk Density of TMC achieved a lower density of 2.5% B_4_C, 7.5% SiC, 12.5% ZrO_2_, and 77.5% Ti composition and developed a minimum density of 2.33 g/cm^3^ and a 24% up to 47% reduced density according to the law of mixture. Additionally, an average reduction of 38% density in the synthesized TMC weight in the final product. Then, the Actual Density of TMC achieved a lower density of 7.5% B_4_C, 7.5% SiC, 2.5% ZrO_2_, and 82.5%Ti composition and developed a minimum density of 2.47 g/cm^3^ and a 16% up to 44% reduced density according to the law of mixture. Additionally, there was an average reduction of 32% density in the synthesized TMC weight in the final product.

Therefore, the Theoretical Density of TMC achieved a lower density of 5% B_4_C, 5% SiC, 5% ZrO_2_, and 85% Ti composition and developed a minimum density value of 0.7 g/cm^3^ and a total of 79% up to 85% reduction in density according to the law of mixture. Additionally, there was an average reduction of 72% density in the synthesized TMC weight in the final product. Then, the relative density was calculated by the ratio of the actual density to the theoretical/calculated density. The high relative density of the sintered specimens indicates the strong interface bonding between the constituent particles with negligible porosity or cavities and the synthesized TMC of composite 5% B_4_C, 5% SiC, 5% ZrO_2_, 85% Ti with a rating of 5.17 gm/cm^3^.

As a result shown on Figure 5, the density weight percentage of the synthesized TMC was significantly reduced, and the weight of the material was also significantly reduced, making the developed material light. The purpose of facilitating weight reduction is the reinforcement in low density engineering materials. Due to this reason, the result, TMC’s actual density was reduced. According to the law of mixture shown on Figure 6, 7.5% B4C, 7.5% SiC, 2.5% ZrO_2_, and 82.5%Ti have developed a minimum density of 2.47 g/cm^3^ and a total density reduction of 16 percent to 44 percent. Additionally, the average density of synthesized TMC weight of the final product is reduced by 32%.

### 3.5. Porosity Measurement

The Archimedes concept was rummage-sale to compute the extent of porosity in TMCs sintered at different temperatures. Sintered dry weight/weight in air (Wd) was measured using a precision balance for each sintered Tmc sample. After that, the fabricated samples were immersed in water and boiled for two hours before being soaked for another 24 h. Suspension weight in water (Ww) of TMC samples was determined. The fabricated sample was soaked and the weight was measured when the water is removed by using dry tissue paper to remove excess water (Ws). The porosity was measured in accordance with Archimedes and determined by calculating using the equation [58]:
(4)$$\varphi =\frac{\mathrm{Ws}-\mathrm{Wd}\;}{\mathrm{Ws}-\mathrm{Ww}}\times 100\%$$
where: φ = porosity (%), Ws = mass of the sample after soaking in distilled water for 24 h (g), Wd = mass of sintered dried sample (g), Ww = mass of sample hanging or suspension in water (g). According to [1], the porosity from theoretical density Equation (4) is being used to evaluate the actual density of each material; hence, Equation (4) is applied to compute the porosity of each material and the accurate porosity is calculated. Variation of Porosity calculated by Archimedes’ principles as given in Equation (5)
(5)$$P=\frac{1-\rho t}{\;\rho a}$$
where P represents the porosity occurring in the material, ρa represents its actual density and ρt represents its theoretical density.

The void content volume was then computed using Agarwal and Broutman’s equation [59,60]. Figure 7 below depicts the porosity of the fabricated samples.

Porosity was calculated using Archimedes’ principles with the minimum porosity of 2.34% has been observed for sample having composition of 7.5%, B_4_C, 7.5%, SiC, 2.5%, ZrO_2_, and 82.5%, Ti and the maximum porosity of 19.5% has been observed for the sample having composition of 5% B_4_C, 5% SiC, 5% ZrO_2_, 85% Ti. For the porosity calculation, the minimum porosity was computed using the theoretical density method rather than calculating porosity by the Archimedes principle. The void content of the synthesized TMC in nine samples was below one; this is indicated that the synthesized TMC are good consolidated engineering materials for the application of automotive and aerospace engineering. The high relative density of the sintered specimens indicates the strong interface bonding between the constituent particles with negligible porosity or cavities.

### 3.6. Estimate the Yield and Tensile Strengths

Since the invention of delamination hardness testing, there have been analyses to approximate other mechanical characteristics from bulk hardness measurements, particularly ultimate tensile strength and yield strength [63]. Hardness analyses have been widely used as a forecasting tool for estimating the yield and tensile strengths of Ni, Fe, Cu, and Al-based alloy systems [13,29,35,36,63,64], as well as nanocrystalline metal systems and metallic glasses, such as titanium [4,65]. Across these various metallic structures, there is an overall interaction for correlating the yield strength, y, and hardness H, $\sigma t=\left(\frac{H}{3}\right)$. This correlation is only acceptable for metallic materials with low strain hardening. If the material displays strain hardening, then the hardness estimation induces strain hardening, and the subsequent hardness assessment is representative of the strain-hardened material rather than the material prior to the measurement [66]. Theoretical equations were developed for equating the tensile and yield strengths to the hardness of metals that strain harden, such as steel, nickel, aluminum, and copper alloys [66]. It has been discovered that the strain induced by a Vickers indenter ranges between 8% and 10% and that the equivalent stress at this strain is approximately Hv/2.9 for steel and Hv/3 for copper alloys. If a metal has power–law strain hardening, the true stress, t, as a function of true strain, can be expressed as follows $\sigma t=KꜪ{\;}^{n}$, where K is the material’s strength coefficient and n is the strain hardening exponent, Tabor developed a relationship equating the tensile stress, UTS, to the Vickers hardness using the approximate stress observed for steels and copper alloys at a strain of 8%. Cahoon et al. [35] improved and simplified this relationship by doing the following,
(6)$$\sigma \mathrm{UTS}=\left(\frac{\mathrm{Hv}}{2.9}\right){\left(\frac{n}{0.217}\right)}^{n}$$

It also discovered a link between the 0.2 percent offset yield strength, y, and the Vickers hardness for a metal with power–law strain hardening behavior. K can be calculated from Equation (3) by assuming that Hv/3 is the stress at a strain of 8%, and the yield strength to Vickers hardness relationship can be described as follows
(7)$$\sigma y=\left(\frac{\mathrm{Hv}}{3}\right){\left(\frac{Ꜫy}{0.08}\right)}^{n}$$
where Ꜫy denotes the true strain at 0.2% offset yield strength. Cahoon et al. [35] determined empirically that Ꜫy was approximately 0.008 for both aluminum and steel samples. Using the following relationship [35], the yield strength can be equated to the Vickers hardness assuming that y can be treated as a constant for all metals
(8)$$\;\sigma y=\left(\frac{\mathrm{Hv}}{3}\right){\left(0.1\right)}^{n}$$

However, the strain-hardening exponent, n, may not be known for a particular material, and obtaining n typically requires direct measurement through tensile testing. Thus, using these relationships as a predictive tool may not be practical for PM processes where the strain hardening behavior is not well characterized. Instead of relying on relationships empirically derived using n, a linear correlation between the strength and hardness can also be used as a predictive tool when the strain hardening behavior is unknown [29]. Despite varying strain hardening behaviors for various types of steels, the Vickers hardness still exhibits a strong linear correlation with the resulting tensile and yield strengths as can be seen from Table 6. However, the strain hardening does have an impact on the strength-to-hardness relationship, and the predicted strengths tend to be lower than the observed strengths for steels exhibiting a large amount of strain hardening [53,54,67]. For a given yield or tensile strength, the hardness values measured were higher than predicted by Equations (4) and (6) with *n* < 0.1. The data trend, however, seemed to follow a similar slope expected from the empirical models with n between 0.05 and 0.1 [68].

Currently, the only available strength-to-hardness correlation for Ti-6Al-4V is an empirical relationship, developed by [64,69] fitting of the Vickers hardness, Hv, and tensile strength, σUTS, for investment cast Ti-6Al-4V components. The tensile test is one of the most important mechanical property evaluation tests. Tensile tests are used for a variety of purposes. Tensile properties are frequently included in material specifications when selecting materials for engineering applications and ensuring quality. The strength of a material is frequently the most important factor to consider [70,71]. Although hardness is commonly used to predict strength in steel and other common alloy systems, titanium works similarly well while adhering to the ASTM standard. Variation of Approximation of computed hardness and tensile Strength is shown in Figure 8.

## 4. Conclusions

The use of TMC is increasing in not only the aerospace and automobile industries but also in marine, biomedical, electronic, chemical, and petrochemical industries. TMC was prepared by the metallurgical powdering technique, which was a low-cost efficient method. The different mechanical properties of the titanium composites were studied as the reinforcement particles obtained in the composites with proper ratios. Both industrial and academic researchers have displayed their interest in TMCs because it has been observed due to the following conclusions, that through the variation of sintering temperature of TMC, the increasing sintering temperature caused the decrease in density and porosity values. The TiO_2_ sample, which is sintered at over 900 °C tends to produce a rutile phase. The addition of Boron carbide and silicon carbide in titanium at the ratio of 2.5% to 12% of TMC has been found to reduce the density of the composite which was helpful to reduce the final product weight. The hardness of TMC showed the best results when B_4_C, SiC, and ZrO_2_ were reinforced with 12% B_4_C, 12.5% SiC, 7.5% ZrO_2_, and 77.5% Ti and were a maximum of 59 in the Rockwell type “C” HRB scale. Hardness increases with the increase in B_4_C, SiC, and ZrO_2_ but decreases with the decreases in B_4_C, SiC, and ZrO_2_.To obtain optimum hardness, the reinforced material can be used in proper proportions and nanoparticles; this is the main result for achieving the best mechanical properties. The reinforcing matrix element, which increased with SiC was found to be very negligible in the pores when the mixture was conducted properly. Apart from the mechanical properties, the XRD pattern showed the matrix at different intensities where the interfacial bonding of the matrix directly affects the strength of the composite.

## Figures and Tables

**Figure 1 materials-15-05525-f001:**
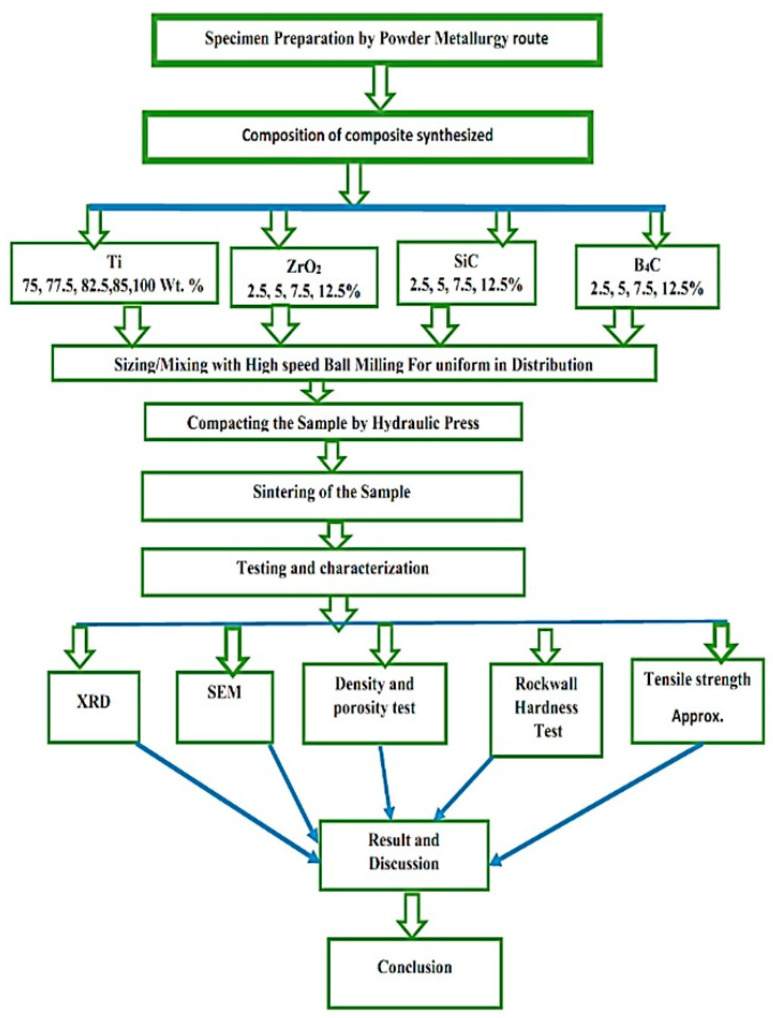
Diagram of TMCs development and characterization flow chart.

**Figure 2 materials-15-05525-f002:**
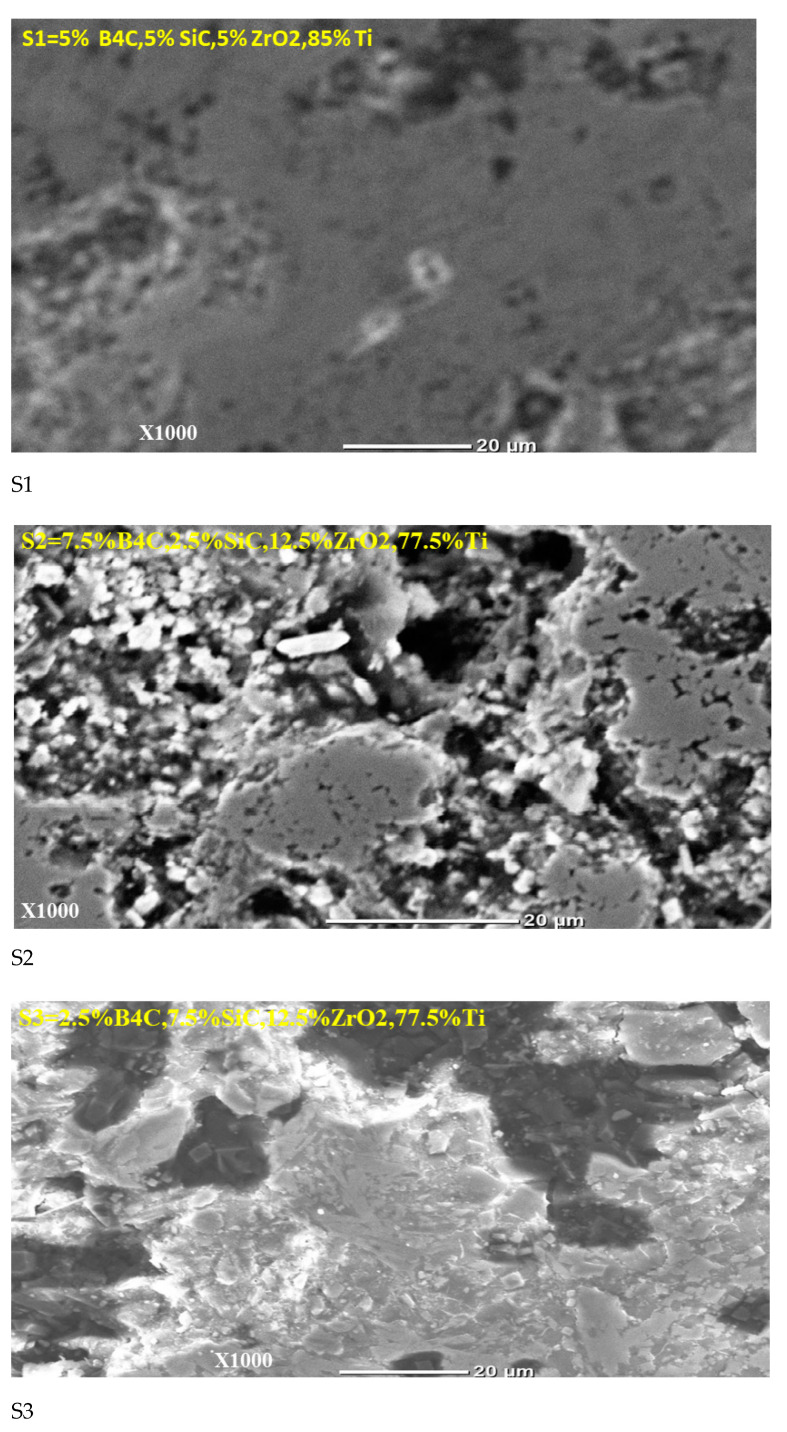
SEM microstructures of machined synthesized TMMC with samples (1) S1, (2) S2, (3) S3, (4) S4, (5) S5, (6) S6, (7) S7, (8) S10.

**Figure 3 materials-15-05525-f003:**
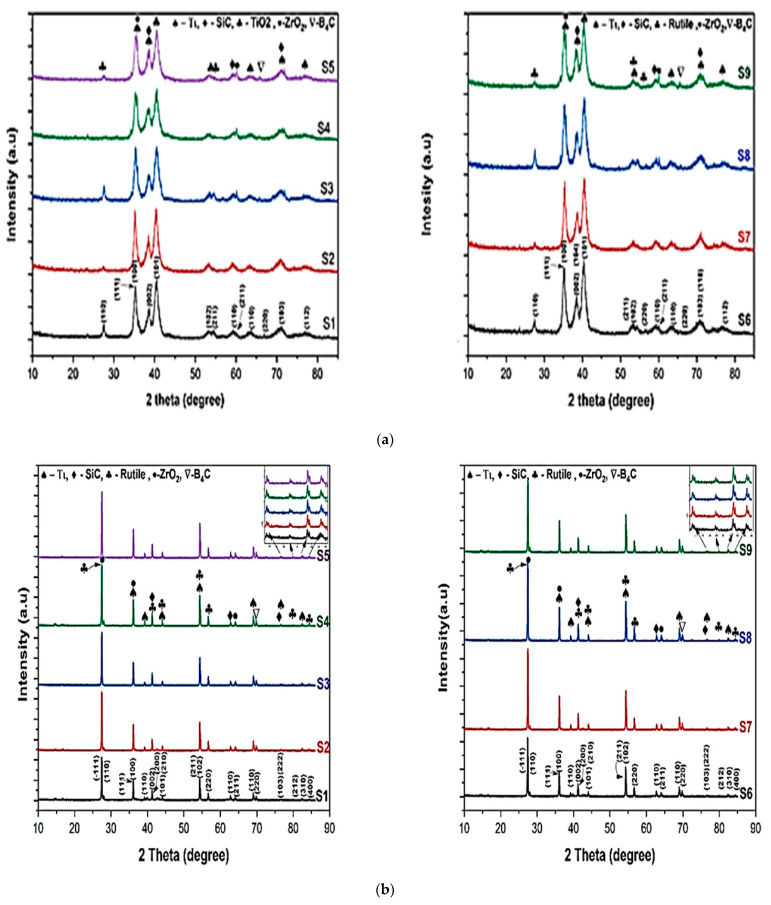
The XRD graph of (**a**) before sintering (**b**) after sintering.

**Figure 7 materials-15-05525-f007:**
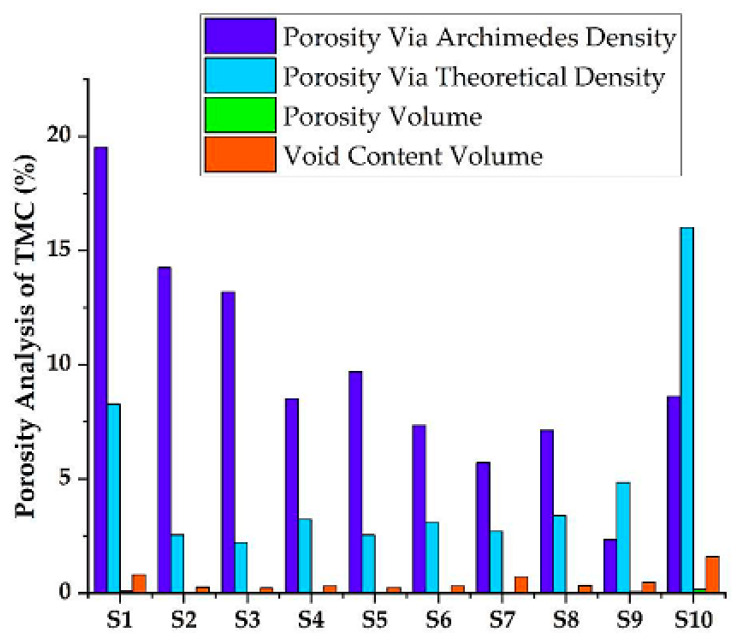
Porosity analysis of synthesized TMC.

**Figure 8 materials-15-05525-f008:**
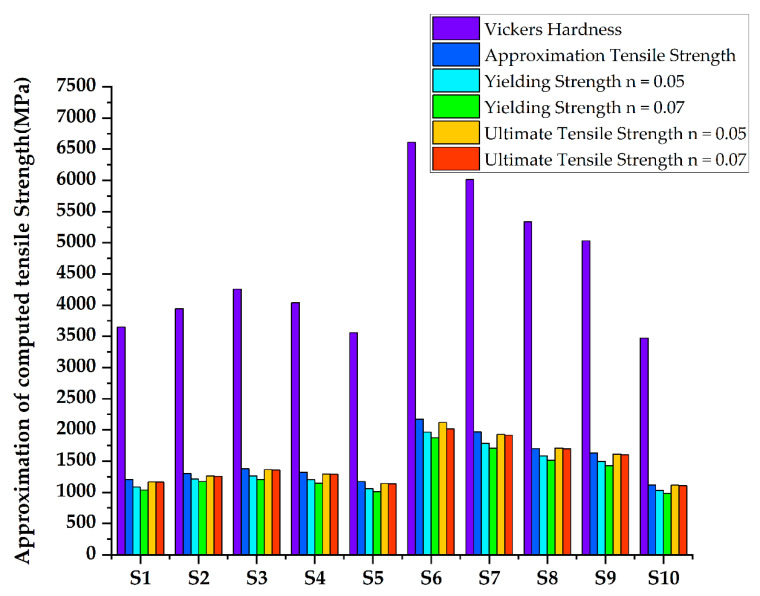
Approximation of computed hardness and tensile Strength.

**Table 1 materials-15-05525-t001:** Sample Notation and Composition.

Sample Notation	Composition
S1	5% B_4_C, 5% SiC, 5% ZrO_2_, 85% Ti
S2	7.5% B_4_C, 2.5% SiC, 12.5% ZrO_2_, 77.5% Ti
S3	2.5% B_4_C, 7.5% SiC, 12.5% ZrO_2_, 77.5% Ti
S4	7.5% B_4_C, 12.5% SiC, 2.5% ZrO_2_, 77.5% Ti
S5	2.5% B_4_C, 12.5% SiC, 7.5% ZrO_2_, 77.5% Ti
S6	5% B_4_C, 7.5% SiC, 12.5% ZrO_2_, 77.5% Ti
S7	5% B_4_C, 12.5% SiC, 7.5% ZrO_2_, 77.5% Ti
S8	5% B_4_C, 7.5% SiC, 2.5% ZrO_2_, 77.5% Ti
S9	7.5% B_4_C, 7.5% SiC, 2.5% ZrO_2_, 77.5% Ti
S10	100% Cp Titanium

**Table 2 materials-15-05525-t002:** Experimental procedure of Titanium MMM synthesized.

No	Sample Symbol	Milling Time (h)	Compressing Pressure (MPa)	Compression Time (min)	Sintering Temperature (°C)	Sintering Time (h)
1	S1	2	50	30	900	2
2	S2	2	50	30	950	2
3	S3	2	50	30	1000	2
4	S4	2	50	30	1050	2
5	S5	2	50	30	1100	2
6	S6	2	50	30	1150	2
7	S7	2	50	30	1200	2
8	S8	2	50	30	1250	2
9	S9	2	50	30	1300	2
10	S10	2	50	30	1150	2

**Table 3 materials-15-05525-t003:** TMC surface micro hardiness through Rockwell hardness “C” type tester.

No.	Sample Symbol	Rockwell Type “C” Micro Hardness (HRB)			
		Trial-1	Trial-2	Trial-3	Average Rockwell Hardness
1	S1	38.5	37.9	38.2	38.2
2	S2	40.9	41.6	41.3	41.3
3	S3	42.9	43.7	42.6	43.1
4	S4	41.7	41.8	42.6	42
5	S5	36.9	36.8	37.4	37
6	S6	59.5	58.5	59	59
7	S7	56.6	55.8	56.1	56.2
8	S8	52.9	51.7	51.8	52.1
9	S9	49.9	49.8	50.8	50.2
10	S10	34.7	35.8	34.9	35.1

**Table 4 materials-15-05525-t004:** Samples Wt. % reinforcement and base metal by law/rule/ of mixture [42].

No		Row Material Powder	Density (ρ) in gm/cm^3^	Weight Percentages (%) in Mixture (x)	Rule of Mixture (ρ × x) in gm/cm^3^
1	S1	B_4_C	2.51	5	0.1255
		SiC	3.21	5	0.1605
		ZrO_2_	5.68	5	0.284
		Ti	4.6	85	3.91
		100%	16	100	4.48
2	S2	B_4_C	2.51	7.5	0.18825
		SiC	3.21	2.5	0.08025
		ZrO_2_	5.68	12.5	0.71
		Ti	4.6	77.5	3.565
		100%	16	100	4.5435
3	S3	B_4_C	2.51	2.5	0.06275
		SiC	3.21	7.5	0.24075
		ZrO_2_	5.68	12.5	0.71
		Ti	4.6	77.5	3.565
		100%	16	100	4.5785
4	S4	B_4_C	2.51	7.5	0.18825
		SiC	3.21	12.5	0.40125
		ZrO_2_	5.68	2.5	0.142
		Ti	4.6	77.5	3.565
		100%	16	100	4.2965
5	S5	B_4_C	2.51	2.5	0.06275
		SiC	3.21	12.5	0.40125
		ZrO_2_	5.68	7.5	0.426
		Ti	4.6	77.5	3.565
		100%	16	100	4.455
6	S6	B_4_C	2.51	5	0.1255
		SiC	3.21	7.5	0.24075
		ZrO_2_	5.68	12.5	0.71
		Ti	4.6	75	3.45
		100%	16	100	4.40275
7	S7	B_4_C	2.51	5	0.1255
		SiC	3.21	12.5	0.71
		ZrO_2_	5.68	7.5	0.24075
		Ti	4.6	75	3.45
		100%	16	100	4.52625
8	S8	B_4_C	2.51	5	0.1255
		SiC	3.21	7.5	0.24075
		ZrO_2_	5.68	2.5	0.142
		Ti	4.6	85	3.91
		100%	16	100	4.41825
9	S9	B_4_C	2.51	7.5	0.18825
		SiC	3.21	7.5	0.24075
		ZrO_2_	5.68	2.5	0.142
		Ti	4.6	82.5	3.795
		100%	16	100	4.366
10	S10	Ti	4.6	100	4.6
		100%	4.6	100	4.6

**Table 6 materials-15-05525-t006:** TMC synthesized by PM approximation of ultimate tensile strength and yielding strength.

No	Sample Symbol	Vickers Hardness (VH)	Vickers Hardness (Mpa)	Approximation Tensile Strength (MPa)	Yielding Strength *n* = 0.05 (MPa)	Yielding Strength *n* = 0.07 (MPa)	Ultimate Tensile Strength *n* = 0.05 (MPa)	Ultimate Tensile Strength *n* = 0.07 (MPa)
1	S1	372	3648	1206.58	1083.76	1034.98	1168.9	1162.15
2	S2	402	3942	1299.66	1211.5	1171	1263	1255.8
3	S3	434	4256	1375.50	1264.4	1207.5	1363.7	1355.8
4	S4	412	4040	1322.40	1200	1146.2	1294.5	1287
5	S5	363	3560	1169.97	1057.6	1010	1140.7	1134
6	S6	674	6610	2171.84	1963.7	1875	2118	2015.58
7	S7	613	6012	1966.38	1786	1705.68	1926.4	1915.3
8	S8	544	5335	1701.62	1584.9	1513	1709.5	1699.58
9	S9	513	5031	1631.29	1494.4	1427.4	1612	1602.7
10	S10	354	3472	1116.95	1031.5	985	1112.5	1106

## Data Availability

Required Data is Embedded within the Article.

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
