# Peer review of "Effect of Sintering Temperatures, Reinforcement Size on Mechanical Properties and Fortification Mechanisms on the Particle Size Distribution of B4C, SiC and ZrO2 in Titanium Metal Matrix Composites"

_materials, 2022, doi:10.3390/ma15165525_

Round 1

Reviewer 1 Report

The authors write that the material will have applications in the space industry. The introduction section lacks a proper literature citation. 

Incorrectly used designations in the paper e.g.

line 108 -B4C line 124 - B4C 

line 118 ZrO2 , line 124 ZrO2ZrO

2

Errors of this type also apply to other designations and are common throughout the paper.

In Table 2, the authors present variants of tests depending, among others, on the temperature range of 900-1300 degrees celcius. If the material is to be used in the cosmetic industry, tests should also be carried out at negative temperatures below 150°C.

Figure 1 and Figure 2 are labeled 1A and5 B. Figure 5. 5.1 5.2..... Please maintain uniformity.

Figure 6 has two diagrams and only one caption.

The authors state in the title that they will study mechanical properties. In addition to microhardness studies, Young's modulus studies, tensile plots of the specimen are missing. These properties will help characterize the structural material.

Author Response

All changes have been highlighted using yellow color and track changes function

Reviewer 1

Comment 1

“The authors write that the material will have applications in the space industry. The introduction section lacks a proper literature citation.” 

Response

Reconstruct introduction

Introduction

Metal matrix composites (MMCs) are gaining popularity in scientific and industry circles due to their appealing physical, mechanical qualities and have a tremendous potential for use in the automobile and aerospace industries [1]. When compared to conventional and homogeneous metal alloys [2], particle-armored MMCs have superior mechanical properties such as strengthening and stiffening [3-4], hardness [5-6], and fracture toughness [7-8]. As a result, composites have the ability to deliver customized mechanical characteristics, making them appealing for a broad array of applications [9]. To enhance the properties of particulate-armored MMCs, recent studies have focused on decreasing reinforcement particles from micrometric domain towards the sub-nano and micro-metric size scales [10-11]. Several studies identified that decreasing the size/dimension of armored plate particles can increase the strength and minimize stress concentration at the corners of nanoparticle-reinforced composites, resulting in increased work hardenability due to the armored plate dislocation effect [12-15]. Many researchers are intrigued by their uses in aerospace and automotive, chemical industry, biomedical [16], and other industries. In the aforementioned applications, the demand for greater quality materials with improved mechanical properties, tribological [17] and machining [18] properties has increased dramatically. Pure titanium is now the most appealing metallic material for aerospace and vehicle applications due to its low density (4.6g/cm3), excellent strength to weight ratios, and great corrosion resistance. Titanium has poor mechanical characteristics and it is brittle and easily fractures at room temperature. Pure elements lack the ability to withstand force or chemical attack and must be combined with additional components in need to balance physical and mechanical qualities. When numerous components are combined, they form a material having unique properties for each individual component [19-20]. Powder metallurgy/PM/ has the most effective method for producing homogenous composite materials [21-26]. Titanium dioxide can be found in thirty polymorphs, including anatase, brookite, and rutile. Anatase phase is metastable and can be converted to rutile by heating treatment. Rutile mode thermodynamics were much greater stability than brookite and anatase at heating and depressed the room for macrocrystalline systems, but under pellet conditions, the rutile structure is described to be thermodynamically stable. Several types of particles, whiskers, or fiber ceramics that can be used as reinforcement in composites have been proposed in previous works to improve the overall properties of Titanium matrix composites, such as Ti5Si3, SiC, TiO2, Al2O3, TiB, TiC, graphene Nano platelet, Nano diamonds, WC, ZrO2, B4C, MoS2, rare earth oxides, such as La2O3, Y2O3, and Nd2O3 and etc. [27-37] . According to the required application of the materials [38-41]. Among these candidates, B4C, SiC, and ZrO2 to titanium base metal have been considered the best reinforcements due to their good compatibility with matrix alloys, particularly coefficient of friction, micro hardness, wear resistance, corrosion resistance, yield strength, tensile strength,  ultimate tensile strength, tensile failure strength, and compressive failure strength, formability, toughness, and biocompatibility. Titanium matrix composites/TMCs/fortified with higher strength and stiffness, than ceramic particles/whiskers have emerged as one of the most promising materials in the automotive and aerospace industries due to excellent properties with higher specific strength and stiffness, damage tolerance, and preferable mechanical characteristics at elevated temperatures [42-45]. TMCs, in particular, have found widespread application in the aforementioned areas due to their high specific strength, specific stiffness, and outstanding mechanical characteristics under high heat and temperature [44]. Because of the excellent and remarkable features such as good hardness, low density, high tensile and compressive strength, high toughness, and excellent machinability, particulate reinforcement of B4C, SiC, and ZrO2 reinforced metal composites have been recognized as potential material requirements for such applications, indicating a potential use in a several range of high-stress applications. The primary rationalizations for using titanium in the aerospace sector are weight savings, particularly as a steel potential substitute; space constraints to substitute Al alloys; thermal resistance and efficiency to substitute Al, Ni, steel alloys; higher corrosion resistance to substitute Al and low alloy steels; and composite compatibility to substitute Al alloys[83,84]. As a result, titanium metal matrix is a popular aerospace engineering material. This material is promising candidate and significant in addressing the scarcity and limitations of other super alloy monotonic materials. A study has suggested that lowering the dimension of the reinforcement detritus to 100nm may improve the strength and ductility of synthesized and development MMCs. Despite the enhancement in nanoparticle MMCs, many major questions remain unresolved. To take use of the benefits of smaller reinforcing particles, such as reduced stress concentration, it is necessary to minimize/eliminate nanoparticle agglomerates and establish a homogeneous spatial contribution and spread of individual particles throughout the matrix. Second, it is acknowledged that the structure and chemistry of matrix/reinforcement interfaces have a significant impact on mechanical properties, and cryo-milling reduces interfacial modifications by suppressing diffusion and chemical reactions at cryogenic temperatures and separating reactive nanoparticle powders from the environment. Third, B4C is intriguing because it has a very high hardness at room temperature, which is only slightly lower than those of cubic BN and diamond; at temperatures exceeding 1200°C, its own hardness has been shown to exceed that of diamond. Furthermore, B4C is cheaper and less challenging to create than diamond and cubic BN. These properties, together with its high melting point, low density, and extraordinary chemical inertness, make B4C an excellent reinforcement for a wide range of metals and found that Ti composites containing nano dimension B4C particles had stronger strength and better tensile ductility than those with micro dimensions B4C particles [12]. Fourth, zirconium dioxide (ZrO2) has outstanding biomechanical qualities such as fracture strength, toughness, and fatigue resistance, as well as low elasticity module and strength, as well as high wear resistance and bio-compatibility. Powder metallurgy has stated as the method of combining, pressing, and sintering the ingredients of a composite. PM is the most effective production process and method for generating homogeneous composite materials. This approach produces exceptional characteristics by achieving good uniformity and low porosity. Fifth, SiC reinforcing was chosen as a reinforcement due to its corrosion resistance, high strength, outstanding thermal stability, formability, ductility, stiffness, low cost, and other characteristics. In this study, Powder Metallurgy was utilized to synthesize Ti-B4C, SiC, and ZrO2 nanocomposites. The mechanical characteristics of the developed TMCs materials' microstructure, densification, micro hardness, sintering temperature influence, and reinforcement size dependence and distribution have all been investigated.

Comment 2

“Line 108 -B4C line 124 - B4C 

Line 118 ZrO2, line 124 ZrO2ZrO2’’

Response

B4C and ZrO2 are uniformly replaced and corrected as needed in their place.

Comment 3

“In Table 2, the authors present variants of the tests depending, among others, on the temperature range of 900-1300 degrees celcius. If the material is to be used in the cosmetic industry, tests should also be carried out at negative temperatures below 150°C.”

Response

This study does not concentrate on the cosmetics industry. This material will be used in the automotive and aerospace industries. The goal of this research is to create alternative materials for automotive and aerospace applications that can be sintered at temperatures ranging from 900 to 1300 degrees Celsius.

Comment 4

“Figure 1 and Figure 2 are labeled 1A and5 B. Figure 5. 5.1 5.2..... Please maintain uniformity.”

Response

Figures 1A and 2A were removed from the content because they were redundant, and they were also replaced by figure 1.

Figure 1. Diagram of TMCs development and characterization flow chart.

Figure 5 was changed to depict a blow, figure 2.

Figure 2. SEM microstructures of machined synthesized TMMC with samples (1) S1, (2) S2, (3) S3, (4) S4, (5) S5, (6) S6,(7) S7, (10) S10

Microstructure Analysis

It is observed that sample S3 (2.5%B4C,7.5%SiC,12.5%ZrO2,77.5%Ti) by increasing the concentrations of ZrO2, decreasing B4C and SiC particles decreased the porosity and densified the surface of the Ti-based metal matrix. In most places on the surface of Ti-based metal matrix sample S3, the ZrO2 particle agglomeration is seen. Because increasing the ZrO2 particle caused agglomeration. Muharrem Pul et al..... [81] investigate that the addition of ZrO2 particles in the metal matrix composite initiated agglomeration.The microstructure of S3 shows that there is no porous structure between the 77.5 %Ti and 2.5%B4C, 7.5%SiC and 12.5%ZrO2 reinforcing particles and the bonding of the phases is very good. We observed from samples S6 and S7 are the same concentrations of ZrO2 as S3, however, due to the increasing of the concentrations of B4C and SiC, S6 and S7 are more porous microstructure. Harish et al..... [82] report that the porosity of the metal matrix composite materials in microstructure increases with the increase of the particles of reinforcing. Therefore, with the same concentration of ZrO2 and different concentrations of B4C and SiC, samples S3, S6, and S7 are different surface morphology.

Comment 5

“Figure 6 has two diagrams and only one caption.”

Response

3.2. XRD Analysis

The elemental phases present in the manufactured samples were analyzed using XRD in accordance with the XRD working principle: Bragg's law[138]. The XRD was performed on a fully computerized powder x-ray diffractometer (XRD7000 X-RAY DIFFRACTOMETER, SHIMADZU Corporation (Japan)) at 40 KV and 30 mA. The XRD spectrum were generated at a 2 degree angle ranging from 10 to 85 degrees with a 0.02 degree step size. Continuous scanning at 3 degrees per minute for 0.40 seconds. Miller indices (hkl) are used to identify various planes of atoms, and the observed diffraction peaks can be related to the planes of atoms to aid in atomic structure and microstructure analysis. When analyzing XRD data, we look for trends that correspond to crystal structure directionality by analyzing the Miller indices of diffraction peaks. The crystal structure determines the position and intensity of peaks in a diffraction pattern. The fabricated samples were subjected to XRD analysis to determine whether any intermetallic compounds were formed during the sintering process[59]. When the diffractometer is linked to the X'pert data collector software, d' values are displayed directly on the diffraction pattern. These d' values were then used to identify different phases using ASTM X-ray diffraction data cards. To confirm the presence of minor precipitate phases detected by the diffraction pattern, the d' values for different phases were obtained using JCPDS cards included with the software and manually compared with the diffraction pattern of all samples[60].

Figure 3, A and B replaced and changed to the plow figure

a)

b)

Figure 3. The XRD graph of a) mild before sintering b) after sintering

Figure 3 represents the XRD graph of the titanium-based metal matrix composite powders with milling before compaction and sintering were performed. The graph shows that there is a dominance of the titanium matrix peaks, which ascribes that in the milling process there was an undesirable interfacial chemical reaction between the hybrid reinforcements and the matrix with less peak are detected with angles of 2θ=27.60 and 54.480 corresponding to (110) and (211) respectively and the rutile TiO2 (JCPDScard number: 021-1276 ) developed. As shown in Figure 3, a and b, titanium with a hexagonal closed packed crystal structure with a = b = c = 1.587 and ∝= β = γ ≠ 90  with an experimental density of 4.6 g/cm3 can be detected in the titanium metal matrix samples, regardless of whether it is before or after sintering. However, in the composite shown in Figure 3. b, the peaks corresponding to distinct phases are recognized as Ti, B4C, SiC, and ZrO2. The presence of Ti, B4C, SiC, ZrO2, and rutile (TiO2) in the titanium metal matrix is highly correlated with the presence of Ti, B4C, SiC, and ZrO2 in the titanium metal matrix. The peak of rutile (TiO2) (JCPDS card number: 021-1276) was shown for all samples at 2θ =27.440, 41.60, 44.080, 79.90 and 84.340 corresponding to the (110), (002), (210), (212) and(400) crystallographic plane respectively. The peak of SiC was shown at angles 2θ =41.280, 62.780, and 76.60   corresponding to the (200), (110), and (103) crystallographic plane (JCPDS card number: 049-1623). The peaks of B4C at angles 2θ = 69.840 correspond to the (220) crystallographic planes. The peaks ZrO2  are also shown at angles of 2θ= 27.50 and 64.10 corresponding to (-111) and (211) crystallographic planes. It is observed that after sintering of the metal matrix of all components of the composite detected by x-ray.

Comment 6

“The authors state in the title that they will study the mechanical properties. In addition to micro-hardness studies, Young's modulus study, tensile plots of the specimen are missing. These properties will help characterize the structural material.”

Response

Corrected and discussed further below

3.6. Estimate the Yield And Tensile Strengths

Since the invention of delamination hardness testing, there have been analyses to approximate other mechanical characteristics from bulk hardness measurements, particularly ultimate tensile strength and yield strength[69]. Hardness analyses have been widely used as a forecasting tool for estimating the yield and tensile strengths of Ni, Fe, Cu, and Al-based alloy systems [69,70-74], as well as nanocrystalline metal systems and metallic glasses like titanium also similarly used[75,76]. Including across these various metallic structures, there is an overall interaction for correlating the yield strength, y, and hardness H, . This correlation, but even so, is only acceptable for metallic materials with low strain hardening. If the material displays strain hardening, then the hardness estimation induces strain hardening, and the subsequent hardness assessment is representative of the strain-hardened material rather than the material prior to the measurement[77]. Theoretical equations were developed for equating the tensile and yield strengths to the hardness of metals that strain harden, such as steel, nickel, aluminum, and copper alloys[77]s. It has been discovered that the strain induced by a Vickers indenter ranges between 8% and 10%, and that the equivalent stress at this strain is approximately H V/2.9 for steel and HV/3 for copper alloys. If a metal has power-law strain hardening, the true stress, t, as a function of true strain, can be expressed as follows , where K is the material's strength coefficient and n is the strain hardening exponent Tabor developed a relationship equating the tensile stress, UTS, to the Vickers hardness using the approximate stress observed for steels and copper alloys at a strain of 8%. Cahoon et al..[70]improved and simplified this relationship by doing the following, , It also discovered a link between the 0.2 percent offset yield strength, y, and the Vickers hardness for a metal with power-law strain hardening behavior. K can be calculated from Eq. (3) by assuming that H V/3 is the stress at a strain of 8%, and the yield strength to Vickers hardness relationship can be described as follows where Ꜫy denotes the true strain at 0.2% off set yield strength. Cahoon et al.. [70] determined empirically that Ꜫy was approximately 0.008 for both aluminum and steel samples. Using the following relationship[70], the yield strength can be equated to the Vickers hardness assuming that y can be treated as a constant for all metalsHowever, the strain-hardening exponent, n, may not be known for a particular material, and obtaining n typically requires direct measurement through tensile testing. Thus, using these relationships as a predictive tool may not be practicable for PM processes where the strain hardening behavior is not well characterized. Instead of relying on relationships empirically derived using n, a linear correlation between the strength and hardness can also be used as a predictive tool when the strain hardening behavior is unknown[74]. show that, despite varying strain hardening behaviors for various types of steels, the Vickers hardness still exhibits a strong linear correlation with the resulting tensile and yield strengths as can be seen from Table 6. However, the strain hardening does have an impact on the strength-to--to-hardness relationship, and the predicted strengths tend to be lower than the observed strengths for steels exhibiting a large amount of strain hardening[74]. For a given yield or tensile strength, the hardness values measured were higher than predicted by Eqs. (4) and (6) with n < 0.1. The data trend, however, seemed to follow a similar slope expected from the empirical models with n between 0.05 and 0.1[79].

Table 6. TMC synthesized by PM approximation of ultimate tensile strength and yielding strength.

No

Sample symbol

Vickers hardness

(VH)

Vickers hardness

(Mpa)

Approximation Tensile Strength (MPa)

Yielding strength n=0.05 (MPa)

Yielding strength n=0.07 (MPa)

Ultimate tensile strength n=0.05 (MPa)

Ultimate tensile strength n=0.07 (MPa)

1

S1

372

3648

1,206.58

1083.76

1034.98

1168.9

1162.15

2

S2

402

3942

1,299.66

1211.5

1171

1263

1255.8

3

S3

434

4256

1,375.50

1264.4

1207.5

1363.7

1355.8

4

S4

412

4040

1,322.40

1200

1146.2

1294.5

1287

5

S5

363

3560

1,169.97

1057.6

1010

1140.7

1134

6

S6

674

6610

2,171.84

1963.7

1875

2118

2015.58

7

S7

613

6012

1,966.38

1786

1705.68

1926.4

1915.3

8

S8

544

5335

1,701.62

1584.9

1513

1709.5

1699.58

9

S9

513

5031

1,631.29

1494.4

1427.4

1612

1602.7

10

S10

354

3472

1,116.95

1031.5

985

1112.5

1106

Figure 8. Approximation of computed hardness and tensile Strength

Currently, the only available strength-to--to-hardness correlation for Ti-6Al-4V is an empirical relationship, developed by [ 73,78] fit of the Vickers hardness, HV, and tensile strength, σUTS, for investment cast Ti-6Al-4V components. The tensile test is one of the most important mechanical property evaluation tests. Tensile tests are used for a variety of purposes. Tensile properties are frequently included in material specifications when selecting materials for engineering applications and ensuring quality. The strength of a material is frequently the most important factor to consider. Although hardness is commonly used to predict strength in steel and other common alloy systems, titanium works similarly well while adhering to the ASTM standard.

Reviewer 2 Report

The manuscript is devoted to the study of the effect of sintering temperature and the type of combination of reinforcing ceramic nanoparticles on the change in hardness and residual porosity of titanium matrix composites. The manuscript has serious flaws that need to be corrected.

1) Please read and rewrite the Introduction section carefully. Some of the claims are highly dubious. In addition, it is necessary to bring bibliographic references to a unified form.

2) Experimental procedures concerning the synthesis of composite powders are described in a very fragmentary way. Need to expand and add details.

3) It is not clear what guided the authors when choosing the compositions of the composites.

4) Figure captions in most cases are not informative.

5) It is necessary to provide SEM images of the synthesized composite powders.

6) The data in Fig. 5 indicates a large agglomeration of ceramic particles. This does not allow to talk about a significant improvement in mechanical properties.

7) In Fig. 6, the signatures of the phases should be given.

8) The title of the manuscript needs to be corrected.

9) It is not very legitimate to conclude the change in the yield strength and tensile strength of composites based on data on changes in hardness. Compression tests of bulk composites should at least be carried out.

 After eliminating these shortcomings, the article can be reviewed again. 

Author Response

All changes have been highlighted using yellow color and track changes function

Reviewer 2

Comment 1

“Please read and rewrite the Introduction section carefully. Some of the claims are highly dubious. In addition, it is necessary to bring bibliographic references to a unified form.”

Response

 corrects and replaces "Sun et al., 2018" with [44]

 corrects and replaces "Jiang et al., 2015”" with [12]

 corrects and replaces "Behera et al. 2020”" with [48]

 correct and replaces " Wang et al., 2004” with [1]

The authors write that the material will have applications in the space industry. The introduction section lacks a proper literature citation.” 

Response

Reconstruct introduction

Introduction

Metal matrix composites (MMCs) are gaining popularity in scientific and industry circles due to their appealing physical, mechanical qualities and have a tremendous potential for use in the automobile and aerospace industries [1]. When compared to conventional and homogeneous metal alloys [2], particle-armored MMCs have superior mechanical properties such as strengthening and stiffening [3-4], hardness [5-6], and fracture toughness [7-8]. As a result, composites have the ability to deliver customized mechanical characteristics, making them appealing for a broad array of applications [9]. To enhance the properties of particulate-armored MMCs, recent studies have focused on decreasing reinforcement particles from micrometric domain towards the sub-nano and micro-metric size scales [10-11]. Several studies identified that decreasing the size/dimension of armored plate particles can increase the strength and minimize stress concentration at the corners of nanoparticle-reinforced composites, resulting in increased work hardenability due to the armored plate dislocation effect [12-15]. Many researchers are intrigued by their uses in aerospace and automotive, chemical industry, biomedical [16], and other industries. In the aforementioned applications, the demand for greater quality materials with improved mechanical properties, tribological [17] and machining [18] properties has increased dramatically. Pure titanium is now the most appealing metallic material for aerospace and vehicle applications due to its low density (4.6g/cm3), excellent strength to weight ratios, and great corrosion resistance. Titanium has poor mechanical characteristics and it is brittle and easily fractures at room temperature. Pure elements lack the ability to withstand force or chemical attack and must be combined with additional components in need to balance physical and mechanical qualities. When numerous components are combined, they form a material having unique properties for each individual component [19-20]. Powder metallurgy/PM/ has the most effective method for producing homogenous composite materials [21-26]. Titanium dioxide can be found in thirty polymorphs, including anatase, brookite, and rutile. Anatase phase is metastable and can be converted to rutile by heating treatment. Rutile mode thermodynamics were much greater stability than brookite and anatase at heating and depressed the room for macrocrystalline systems, but Under pellet conditions, the rutile structure is described to be thermodynamically stable. Several types of particles, whiskers, or fiber ceramics that can be used as reinforcement in composites have been proposed in previous works to improve the overall properties of Titanium matrix composites, such as Ti5Si3, SiC, TiO2, Al2O3, TiB, TiC, graphene Nano platelet, Nano diamonds, WC, ZrO2, B4C, MoS2, rare earth oxides, such as La2O3, Y2O3, and Nd2O3 and etc. [27-37] . According to the required application of the materials [38-41]. Among these candidates, B4C, SiC, and ZrO2 to titanium base metal have been considered the best reinforcements due to their good compatibility with matrix alloys, particularly coefficient of friction, micro hardness, wear resistance, corrosion resistance, yield strength, tensile strength,  ultimate tensile strength, tensile failure strength, and compressive failure strength, formability, toughness, and biocompatibility. Titanium matrix composites/TMCs/fortified with higher strength and stiffness, than ceramic particles/whiskers have emerged as one of the most promising materials in the automotive and aerospace industries due to excellent properties with higher specific strength and stiffness, damage tolerance, and preferable mechanical characteristics at elevated temperatures [42-45]. TMCs, in particular, have found widespread application in the aforementioned areas due to their high specific strength, specific stiffness, and outstanding mechanical characteristics under high heat and temperature [44]. Because of the excellent and remarkable features such as good hardness, low density, high tensile and compressive strength, high toughness, and excellent machinability, Particulate reinforcement of B4C, SiC, and ZrO2 reinforced metal composites have been recognized as potential material requirements for such applications, indicating a potential use in a several range of high-stress applications. The primary rationalizations for using titanium in the aerospace sector are weight savings, particularly as a steel potential substitute; space constraints to substitute Al alloys; thermal resistance and efficiency to substitute Al, Ni, steel alloys; higher corrosion resistance to substitute Al and low alloy steels; and composite compatibility to substitute Al alloys[83-85]. As a result, titanium metal matrix is a popular aerospace engineering material. This material is promising candidate and significant in addressing the scarcity and limitations of other super alloy monotonic materials. A study has suggested that lowering the dimension of the reinforcement detritus to 100nm may improve the strength and ductility of synthesized and development MMCs. Despite the enhancement in nanoparticle MMCs, many major questions remain unresolved. To take use of the benefits of smaller reinforcing particles, such as reduced stress concentration, it is necessary to minimize/eliminate nanoparticle agglomerates and establish a homogeneous spatial contribution and spread of individual particles throughout the matrix. Second, it is acknowledged that the structure and chemistry of matrix/reinforcement interfaces have a significant impact on mechanical properties, and cryo-milling reduces interfacial modifications by suppressing diffusion and chemical reactions at cryogenic temperatures and separating reactive nanoparticle powders from the environment. Third, B4C is intriguing because it has a very high hardness at room temperature, which is only slightly lower than those of cubic BN and diamond; at temperatures exceeding 1200°C, its own hardness has been shown to exceed that of diamond. Furthermore, B4C is cheaper and less challenging to create than diamond and cubic BN. These properties, together with its high melting point, low density, and extraordinary chemical inertness, make B4C an excellent reinforcement for a wide range of metals and found that Ti composites containing nano dimension B4C particles had stronger strength and better tensile ductility than those with micro dimensions B4C particles [12]. Fourth, zirconium dioxide (ZrO2) has outstanding biomechanical qualities such as fracture strength, toughness, and fatigue resistance, as well as low elasticity module and strength, as well as high wear resistance and bio-compatibility. Powder metallurgy has stated as the method of combining, pressing, and sintering the ingredients of a composite. PM is the most effective production process and method for generating homogeneous composite materials. This approach produces exceptional characteristics by achieving good uniformity and low porosity. Fifth, SiC reinforcing was chosen as a reinforcement due to its corrosion resistance, high strength, outstanding thermal stability, formability, ductility, stiffness, low cost, and other characteristics. In this study, Powder Metallurgy was utilized to synthesize Ti-B4C, SiC, and ZrO2 nanocomposites. The mechanical characteristics of the developed TMCs materials' microstructure, densification, micro hardness, sintering temperature influence, and reinforcement size dependence and distribution have all been investigated.

Comment 2

“Experimental procedures concerning the synthesis of composite powders are described in a very fragmentary way. Need to expand and add details.”

Response

  • Synthesis Titanium Metal Matrix Reinforced with Nano Particles by Powder Metallurgy process

Powder metallurgy/PM/ is the best promising method for producing TMCs. Despite being a more expensive technique, it has the advantage of creating precision components without melting. The science of producing metal powders/particles and finished/semifinished items from mixed/alloyed powders with/without nonmetallic elements is known as powder metallurgy. Powder metallurgy consists of three distinct steps: (A) the combination of metal and reinforcing powders, (B) powder compaction/squeezing to form a green material body, and (C) sintering, which is frequently followed by additional processing.

  • Mixing or Blending and Sizing of Powders

Blending is the procedure of combining/unifying powders/particles that have different particle/powder sizes and shapes by passing them through the same simple mechanism. Blending should be used to achieve a consistent distribution of particle sizes and reduce porosity [52,53]. High-speed dry ball milling was used to grind/blend  Ti, SiC, ZrO2, and B4C nanoparticle powders. The ball mill was outfitted with a high-speed spindle and ran for two hours to produce a homogeneous powder mixture. The powders were then obtained with the desired grain size and appropriate for the subsequent process. The weight when combined totally of the powders is 5gm. Because these Nano-particles were a uniform particle size of 100m, a powder-to--to-ball weight of 1:5 is ideal for successful mixing and sizing for 2 hours. The milled powder results of ten (10) samples of Ti-based reinforced composites were synthesized using a high-speed dry ball milling machine.

  • Compaction

Utilizing a proper punch and die to generate green compacts using mechanical or hydraulic presses [52], powder mixtures are widely compacted. The powder combinations were cold crushing, squeezing, and compacting at an appropriate pressure with the use of a uniaxial press [54]. Compaction has stated as the development procedures, system, and methods occurring in compacting and squeezing metallic particles in a hydraulically driven due to the required shape. This hydraulic presses have a owing capacity of 25 tons, a 200mm diameter pressure plate, and a 150mm ram stroke. To produce the specimen, milled metallic and ceramic powders are introduced into the die's cavity. For 30 minutes, all prepared samples are compressed at 50 mpa. After being formed at room temperature, the product is a specimen discharged from the die cavity at room temperature. In this experiment, hydraulic press samples were used to condense the milling power. Following completion with a hydraulic cylindrical compression lower type bucket elevator and a green compacted sample shape, the cylindrical shaped specimens were visible.

  • Sintering

Sintering is defined as the procedure, method, and system for binding particles together by heating green compacts in a controlled environment. To sinter materials below their melting points, mesh belt, walking beam, pusher type, and batch furnaces are all used [52]. Many researchers assert that the highest sintering temperature is used to produce components with good surface finish and quality, and it has been demonstrated that as the sintering temperature rises, so do the material's mechanical properties [55,56]. The titanium sintering process was kept at a temperature ranging from 750 to 1350°C, according to [57,58]. Powder metallurgy operates in the solid state below the melting point in a material with a ratio greater than 0.5 times the melting temperature and less than 0.8 times the melting temperature. Sintering is the compacting and formation of sample solids through heating in a vacuum furnace. The compression rates for samples one through ten sintered in the vacuum box furnace for 2 hours at room temperature were 9000C, 9500C, 10000C, 10500C, 11000C,11500C,12000C,12500C,13000C, and 13500C, respectively.

Comment 3

“It is not clear what guided the authors when choosing the composition of the composites.”

Response

Composites are typically composed of two and more than two main constituents. The advancement of conceptual and theoretical framework techniques that lead to the selection of constituents for composites that attempt to maximize one or more aspects of thermomechanical achievement and ability. There are six fundamental concepts that guide the composition's selection: (1) defining the material for engineering performance using performance or "significance advantage" index values; (2) developing material property statistics from which the indicators can be plotted; and (3) defining boundaries to describe the package of properties that a given composite scheme could conceivably satisfy on the rankings. (4) Application domains and disciplines: (5) the cost of exporting and synthesizing the composite matrix and reinforcements and (6) reinforcement material classification and type 

Comment 4

“Figure captions in most cases are not informative.”

Response

All figure captions that are not informative in most cases are removed and corrected.

Comment 5 and 6

“5. It is necessary to provide SEM images of the synthesized composite powders.

  1. The data in Fig. 5 indicates a large agglomeration of ceramic particles. This does not allow talking about a significant improvement in mechanical properties.”

Figure 5 was changed to depict a blow, figure 2.

Figure 5 was changed to depict a blow, figure 2.

Figure 2. SEM microstructures of machined synthesized TMMC with samples (1) S1, (2) S2, (3) S3, (4) S4, (5) S5, (6) S6,(7) S7, (10) S10

Microstructure Analysis

It is observed that sample S3 (2.5%B4C,7.5%SiC,12.5%ZrO2,77.5%Ti) by increasing the concentrations of ZrO2, decreasing B4C and SiC particles decreased the porosity and densified the surface of the Ti-based metal matrix. In most places on the surface of Ti-based metal matrix sample S3, the ZrO2 particle agglomeration is seen. Because increasing the ZrO2 particle caused agglomeration. Muharrem Pul et al..... [81] investigate that the addition of ZrO2 particles in the metal matrix composite initiated agglomeration.The microstructure of S3 shows that there is no porous structure between the 77.5 %Ti and 2.5%B4C, 7.5%SiC and 12.5%ZrO2 reinforcing particles and the bonding of the phases is very good. We observed from samples S6 and S7 are the same concentrations of ZrO2 as S3, however, due to the increasing of the concentrations of B4C and SiC, S6 and S7 are more porous microstructure. Harish et al..... [82] report that the porosity of the metal matrix composite materials in microstructure increases with the increase of the particles of reinforcing. Therefore, with the same concentration of ZrO2 and different concentrations of B4C and SiC, samples S3, S6, and S7 are different surface morphology.

Comment 7

“Figure 6 has two diagrams and only one caption.”

Response

3.2. XRD Analysis

The elemental phases present in the manufactured samples were analyzed using XRD in accordance with the XRD working principle: Bragg's law[138]. The XRD was performed on a fully computerized powder x-ray diffractometer (XRD7000 X-RAY DIFFRACTOMETER, SHIMADZU Corporation (Japan)) at 40 KV and 30 mA. The XRD spectrum were generated at a 2 degree angle ranging from 10 to 85 degrees with a 0.02 degree step size. Continuous scanning at 3 degrees per minute for 0.40 seconds. Miller indices (hkl) are used to identify various planes of atoms, and the observed diffraction peaks can be related to the planes of atoms to aid in atomic structure and microstructure analysis. When analyzing XRD data, we look for trends that correspond to crystal structure directionality by analyzing the miller indices of diffraction peaks. The crystal structure determines the position and intensity of peaks in a diffraction pattern. The fabricated samples were subjected to XRD analysis to determine whether any intermetallic compounds were formed during the sintering process[59]. When the diffractometer is linked to the X'pert data collector software, d' values are displayed directly on the diffraction pattern. These d' values were then used to identify different phases using ASTM X-ray diffraction data cards. To confirm the presence of minor precipitate phases detected by the diffraction pattern, the d' values for different phases were obtained using JCPDS cards included with the software and manually compared with the diffraction pattern of all samples[60].

Figure 3, A and B replaced and changed to the plow figure

a)

b)

Figure 3. The XRD graph of a) mild before sintering b) after sintering

Figure 3 represents the XRD graph of the titanium-based metal matrix composite powders with milling before compaction and sintering were performed. The graph shows that there is a dominance of the titanium matrix peaks, which ascribes that in the milling process there was an undesirable interfacial chemical reaction between the hybrid reinforcements and the matrix with less peak are detected with angles of 2θ=27.60 and 54.480 corresponding to (110) and (211) respectively and the rutile TiO2 (JCPDScard number: 021-1276 ) developed. As shown in Figure 3, a and b, titanium with a hexagonal closed packed crystal structure with a = b = c = 1.587 and ∝= β = γ ≠ 90  with an experimental density of 4.6 g/cm3 can be detected in the titanium metal matrix samples, regardless of whether it is before or after sintering. However, in the composite shown in Figure 3. b, the peaks corresponding to distinct phases are recognized as Ti, B4C, SiC, and ZrO2. The presence of Ti, B4C, SiC, ZrO2, and rutile (TiO2) in the titanium metal matrix is highly correlated with the presence of Ti, B4C, SiC, and ZrO2 in the titanium metal matrix. The peak of rutile (TiO2) (JCPDS card number: 021-1276) was shown for all samples at 2θ =27.440, 41.60, 44.080, 79.90 and 84.340 corresponding to the (110), (002), (210), (212) and(400) crystallographic plane respectively. The peak of SiC was shown at angles 2θ =41.280, 62.780, and 76.60   corresponding to the (200), (110), and (103) crystallographic plane (JCPDS card number: 049-1623). The peaks of B4C at angles 2θ = 69.840 correspond to the (220) crystallographic planes. The peaks ZrO2  are also shown at angles of 2θ= 27.50 and 64.10 corresponding to (-111) and (211) crystallographic planes. It is observed that after sintering of the metal matrix of all components of the composite detected by x-ray.

Comment 8

The title of the manuscript needs to be corrected.

Response: Revision is done as per suggestions

Comment 9

It is not very legitimate to conclude the change in the yield strength and tensile strength of composites based on data on changes in hardness. Compression tests of bulk composites should at least be carried out.

Response: Many thanks for suggestions; compression tests falls outside the scope of journa

Reviewer 3 Report

- Sentences about the experiment should be moved to the section "Experimental procedure and work", eg "The surface topography of the machined surface was investigated using a scanning 177 electron microscopy / SEM, Model JCM / 6000PLUS BENCH TOP SEM, Jeol Jap".

- XRD results are practically not discussed, description should be completed. The figures do not show which diffractogram corresponds to which sample.

- More experimental data should be provided, e.g. on XRD (exposure time, measuring step, surface preparation)

- I believe that figures 1 to 4 are redundant, possibly you can leave figure 3

- The authors should edit the text and standardize the nomenclature of the samples (A ... or S .. or MS ..)

- Why was SEM performed for selected samples? Why is the quality of figure 5.1 so poor?

- The authors included unnecessary figures 1-4 and they do not show photos of the microhardness. Why were only 3 measurements made for each sample?

- Figures 7 to 14 have no physical sense. The composition of the tested samples is not monotonic (e.g. addition of one other component or % increase in the component content). The tested samples have a completely different chemical composition and it is impossible to compare their properties in the given way. The authors could collate the results in one or two graphs. In this way, it is possible to show the relationship between the chemical composition and hardness or density. In the present state, the reader has a difficult task to clearly state the influence of the chemical composition on the properties. Only in conclusions do we find out which samples are the most promising.

I believe that the manuscript is chaotically written, the results are intertwined with elements of the experiment. It is also difficult to find the novelty of the results obtained. Although the work is quite extensive, in fact it can be said that the authors performed XRD (no description !!!), SEM (for selected samples) and microhardness (3 trials each). In my opinion, the article as it stands cannot be published in Materials.

Author Response

Reviewer 3

Comment1

“Sentences about the experiment should be moved to the section "Experimental procedure and work", eg "The surface topography of the machined surface was investigated using a scanning 177 electron microscopy / SEM Model JCM / 6000PLUS BENCH TOP SEM, Jeol Jap".”

Response

This sentence was removed from the text because it was redundant.

"The surface topography of the machined surface was investigated using a scanning 177 electron microscopy / SEM, Model JCM / 6000PLUS BENCH TOP SEM, Jeol Jap".”

The paragraph has been corrected as follows:

The surface topography of the synthesized specimen was investigated using a SEM scanning to show that the SEM microstructures of the synthesized TMMC and base-Ti6Al4V specimens had a coarse lamellar + microstructure with phase separation created during sintering at high temperature and subsequent slow cooling rate, and Figure 2 shows the XRD Analysis of the synthesized sample [46]. 

Figure 5 was changed to depict a blow, figure 2.

Figure 2. SEM microstructures of machined synthesized TMMC with samples (1) S1, (2) S2, (3) S3, (4) S4, (5) S5, (6) S6,(7) S7, (10) S10

Microstructure Analysis

It is observed that sample S3 (2.5%B4C,7.5%SiC,12.5%ZrO2,77.5%Ti) by increasing the concentrations of ZrO2, decreasing B4C and SiC particles decreased the porosity and densified the surface of the Ti-based metal matrix. In most places on the surface of Ti-based metal matrix sample S3, the ZrO2 particle agglomeration is seen. Because increasing the ZrO2 particle caused agglomeration. Muharrem Pul et al..... [81] investigate that the addition of ZrO2 particles in the metal matrix composite initiated agglomeration.The microstructure of S3 shows that there is no porous structure between the 77.5 %Ti and 2.5%B4C, 7.5%SiC and 12.5%ZrO2 reinforcing particles and the bonding of the phases is very good. We observed from samples S6 and S7 are the same concentrations of ZrO2 as S3, however, due to the increasing of the concentrations of B4C and SiC, S6 and S7 are more porous microstructure. Harish et al..... [82] report that the porosity of the metal matrix composite materials in microstructure increases with the increase of the particles of reinforcing. Therefore, with the same concentration of ZrO2 and different concentrations of B4C and SiC, samples S3, S6, and S7 are different surface morphology.

Comment 2 and 3

“2. XRD results are practically not discussed, the description should be completed. The figures do not show which diffractogram corresponds to which sample.

  1. More experimental data should be provided, e.g., with XRD (exposure time, measuring step, surface preparation)”

Response

Figure 6 has been replaced by Figures 3A and 3B, which are discussed further XRD analysis below.

3.2. XRD Analysis

The elemental phases present in the manufactured samples were analyzed using XRD in accordance with the XRD working principle: Bragg's law[138]. The XRD was performed on a fully computerized powder x-ray diffractometer (XRD7000 X-RAY DIFFRACTOMETER, SHIMADZU Corporation (Japan)) at 40 KV and 30 mA. The XRD spectrum were generated at a 2 degree angle ranging from 10 to 85 degrees with a 0.02 degree step size. Continuous scanning at 3 degrees per minute for 0.40 seconds. Miller indices (hkl) are used to identify various planes of atoms, and observed diffraction peaks can be related to the planes of atoms to aid in atomic structure and microstructure analysis. When analyzing XRD data, we look for trends that correspond to crystal structure directionality by analyzing the miller indices of diffraction peaks. The crystal structure determines the position and intensity of peaks in a diffraction pattern. The fabricated samples were subjected to XRD analysis to determine whether any intermetallic compounds were formed during the sintering process [59]. When the diffractometer is linked to the X'pert data collector software, d' values are displayed directly on the diffraction pattern. These d' values were then used to identify different phases using ASTM X-ray diffraction data cards. To confirm the presence of minor precipitate phases detected by the diffraction pattern, the d' values for different phases were obtained using JCPDS cards included with the software and manually compared with the diffraction pattern of all samples[60].

Figure 3, A and B replaced and changed to the plow figure

a)

b)

Figure 3. The XRD graph of a) mild before sintering b) after sintering

Figure 3 represents the XRD graph of the titanium-based metal matrix composite powders with milling before compaction and sintering were performed. The graph shows that there is a dominance of the titanium matrix peaks, which ascribes that in the milling process there was an undesirable interfacial chemical reaction between the hybrid reinforcements and the matrix with less peak are detected with angles of 2θ=27.60 and 54.480 corresponding to (110) and (211) respectively and the rutile TiO2 (JCPDScard number: 021-1276 ) developed. As shown in Figure 3, a and b, titanium with a hexagonal closed packed crystal structure with a = b = c = 1.587 and ∝= β = γ ≠ 90  with an experimental density of 4.6 g/cm3 can be detected in the titanium metal matrix samples, regardless of whether it is before or after sintering. However, in the composite shown in Figure 3. b, the peaks corresponding to distinct phases are recognized as Ti, B4C, SiC, and ZrO2. The presence of Ti, B4C, SiC, ZrO2, and rutile (TiO2) in the titanium metal matrix is highly correlated with the presence of Ti, B4C, SiC, and ZrO2 in the titanium metal matrix. The peak of rutile (TiO2) (JCPDS card number: 021-1276) was shown for all samples at 2θ =27.440, 41.60, 44.080, 79.90 and 84.340 corresponding to the (110), (002), (210), (212) and(400) crystallographic plane respectively. The peak of SiC was shown at angles 2θ =41.280, 62.780, and 76.60   corresponding to the (200), (110), and (103) crystallographic plane (JCPDS card number: 049-1623). The peaks of B4C at angles 2θ = 69.840 correspond to the (220) crystallographic planes. The peaks ZrO2  are also shown at angles of 2θ= 27.50 and 64.10 corresponding to (-111) and (211) crystallographic planes. It is observed that after sintering of the metal matrix of all components of the composite detected by x-ray.

Comment 4

I believe that Figures 1 to 4 are redundant, possibly you can leave Figure 3

Response

Figures 1A and 2A were removed from the content because they were redundant, and they were also replaced by below  figure 1.

Figure 1. Diagram of TMCs development and characterization flow chart.

Comment 5

The authors should edit the text and standardize the nomenclature of the samples (A ... or S .. or MS ..)

Response

Throughout the study, all symbols are uniformly changed to the S - symbol.

S6 and S7 were exchanged together in the previous composition.

S6= 5%B4C,12.5%SiC,7.5%ZrO2,77.5%Ti

S7= 5%B4C,7.5%SiC,12.5%ZrO2,77.5%Ti

As a result of the composition of corrected S6 and S7 as blow

S6= 5%B4C,7.5%SiC,12.5%ZrO2,77.5%Ti

S7= 5%B4C,12.5%SiC,7.5%ZrO2,77.5%Ti

Comment 6

“Why was SEM performed for selected samples? Why is the quality of Figure 5.1 so poor?”

Response

The scanning electron microscope is one of the most commonly used techniques for analyzing nanomaterials and nanostructures (SEM). A scanning electron microscope (SEM) creates an image by scanning a surface with a focused stream of electrons. Focused electron beam interferes with the specimen, producing a variety of signals that can be used to determine surface properties such as morphology and composition. Furthermore, the microstructure is directly proportional to mechanical and chemical properties. As a result, samples with better mechanical properties were chosen to generalize the findings of this study.

Figure 5.1 has been removed entirely due to the poor quality of the camera captured in Figure 5.1.

Comment 7

“The authors included unnecessary Figures 1-4 and they do not show photos of the micro hardness. Why were only 3 measurements made for each sample?”

Response

Figures 1A and 2A were removed from the content because they were redundant, and they were also replaced by below  figure 1.

Figure 1. Diagram of TMCs development and characterization flow chart.

Repeated trials are when you take multiple measurements of the same thing to improve data reliability. Repeating an experiment several times allows you to determine whether the results were an outlier or the norm. The more trials you complete, the closer your average becomes to the true value. Three (3) trials are sufficient to determine an engineering application's average value.

The indentation of the Rockwell hardness tester image is not included and clearly not visible due to the poor quality of the camera capture.

Comment 8

“Figures 7 to 14 have no physical sense. The composition of the tested samples is not monotonic (e.g., addition of one other component or % increase in the component content). The tested samples have a completely different chemical composition and it is impossible to compare their properties in the given way. The authors could collate the results in one or two graphs. In this way, it is possible to show the relationship between the chemical composition and hardness or density. In the present state, the reader has a difficult task to clearly state the influence of the chemical composition on the properties. Only in the conclusions do we determine which samples are the most promising.”

Response

The content 3.3-3.5 subtitle and Figure 7-14 will be improved as a result of gathering information and summarizing it as assistance and input to the conclusion.

3.3. Surface Hardness testing of TMCs

Rockwell hardness examinations are the most extensively used hardness measuring techniques in the manufacturing sector. Diamond indenters are used to achieve various Rockwell hardness scales currently specified in ISO 6508-1, the most important of which are HRC, HRA, and HRN. The problems in introducing assessment methods to the measuring capabilities of the hardness test machines demonstrate the industry's requirement for more accurate calibration techniques within Rockwell hardness investigation machines. The ASTM E18 and 28 standard testing procedure was used to determine the hardness number of the specimens using the Rockwell hardness tester scale as tabulated in Table 3 . The indenter utilized was with 150kg Brale. The load application time is 15 seconds.[47]. Variation of Rockwell micro hardness is shown in Figure 7.

Table 3. TMC surface micro hardiness through Rockwell hardness ‘’C’’ type tester.

NO

Sample symbol

Rockwell type "C"

Micro hardness (HRB)

Trial -1

Trial -2

Trial -3

Average Rockwell hardness

1

S1

38.5

37.9

38.2

38.2

2

S2

40.9

41.6

41.3

41.3

3

S3

42.9

43.7

42.6

43.1

4

S4

41.7

41.8

42.6

42

5

S5

36.9

36.8

37.4

37

6

S6

59.5

58.5

59

59

7

S7

56.6

55.8

56.1

56.2

8

S8

52.9

51.7

51.8

52.1

9

S9

49.9

49.8

50.8

50.2

10

S10

34.7

35.8

34.9

35.1

Figure 4. Variation of Rockwell micro hardness of machined surface analysis.

According to Table 4, Samples Wt. % reinforcement and base metal by law of mixture, the lower the hardness created in sintered specimens, lower the minimum sintering temperature/heat. As a result, factors such as insufficient reinforcement, particle dispersion, clustering of reinforced particles, temperature mismatch between particles and matrix, and particle size discrepancies between matrixes and reinforcing phases all affect the hardness of such composites. Hardness is caused by thermal mismatch, but clustering and insufficient dispersion can result in a decrease in hardness.

3.4. Density and Porosity Measurement

According to the density of the reinforcing material, the phase and size of the combining components, and the process of manufacturing, the composite material the density can increase or decrease[61]. Archimedes' principle was used to estimate the bulk density, porosity, and water absorption of sintered samples. The specimen's sintered weight was first determined using a precision digital weighing balance (HR-250AZ, A&D Company Limited, Korea). The drop in density can be attributed to the reinforcing particles' decreased density and the creation of porosity[62].The specimen was then immersed in 70°C hot water for 2 hours, and the soaked weight was calculated by ASTME Designations C20 – 00 and [40,62]. After that, to what extent to weight occurring in TMCs compacts and sintered were measured using a tumbler full through water into that the samples were suspended/hang down inside the water.

Table 4. Samples Wt. % reinforcement and base metal by law of mixture [48].

No

Row material powder

Density (ρ) in gm/cm3

Weight percentages (%) in mixture (x)

Rule of mixture (ρ × x) in gm/cm3

1

S1

B4

2.51

5

0.1255

 SiC,

3.21

5

0.1605

ZrO2

5.68

5

0.284

Ti

4.6

85

3.91

100%

16

100

4.48

2

S2

B4

2.51

7.5

0.18825

 SiC,

3.21

2.5

0.08025

ZrO2

5.68

12.5

0.71

Ti

4.6

77.5

3.565

100%

16

100

4.5435

3

S3

B4

2.51

2.5

0.06275

 SiC,

3.21

7.5

0.24075

ZrO2

5.68

12.5

0.71

Ti

4.6

77.5

3.565

100%

16

100

4.5785

4

S4

B4

2.51

7.5

0.18825

 SiC,

3.21

12.5

0.40125

ZrO2

5.68

2.5

0.142

Ti

4.6

77.5

3.565

100%

16

100

4.2965

5

S5

B4

2.51

2.5

0.06275

 SiC,

3.21

12.5

0.40125

ZrO2

5.68

7.5

0.426

Ti

4.6

77.5

3.565

100%

16

100

4.455

6

S6

B4

2.51

5

0.1255

 SiC,

3.21

7.5

0.24075

ZrO2

5.68

12.5

0.71

Ti

4.6

75

3.45

100%

16

100

4.40275

7

S7

B4

2.51

5

0.1255

 SiC,

3.21

12.5

0.71

ZrO2

5.68

7.5

0.24075

Ti

4.6

75

3.45

100%

16

100

4.52625

8

S8

B4

2.51

5

0.1255

 SiC,

3.21

7.5

0.24075

ZrO2

5.68

2.5

0.142

Ti

4.6

85

3.91

100%

16

100

4.41825

9

S9

B4

2.51

7.5

0.18825

 SiC,

3.21

7.5

0.24075

ZrO2

5.68

2.5

0.142

Ti

4.6

82.5

3.795

100%

16

100

4.366

10

S10

Ti

4.6

100

4.6

100%

4.6

100

4.6

Through the Archimedes principle, the weight of sintered, soaked, and submerged materials was examined following equation: [1, 63, 64, 80]

               Eqn (1)

   Eqn (2)

To analyze the generated nanocomposites, the genuinely density of all sintered specimens was tested using the Archimedes method and a density measurement device with a precision digital weighing balance (HR-250AZ, A&D Company limited, Korea). The theoretical density was then computed using Agarwal and Broutman's equation [65-68] given in Table 5,

Eq (3).

Where Wf denotes the weight fraction of reinforcement, Wm is the weight fraction of Ti6Al4V, and the denotes the theoretical density of the composite. ρf represents the density of reinforcements SiC (3.21g/cm3), B4C (2.52 g/cm3), and ZrO2 (5.68 g/cm3); ρm represents the density of the Ti6Al4V matrix (4.43 g/cm3). The variation of bulk density was then computed as illustrated in Figure 8. The high relative density of the sintered specimens implies that the constituent particles have strong interface bonding with negligible porosity or voids.

Table 5:  Different Sintered Samples Density Analysis

No

Sintered

 Sample

composition

Bulk Densityg/cm3

Actual density by Archimedes principles (gm/cm3)

Theoretical density (gm/cm3)

Relative density gm/cm3

Porosity volume%

Porosity by theoretical density %

Porosity by    Archimedes principles %

1

S1

2.91734

3.62416

0.7

5.17

0.08277

8.27777

19.5029714

2

S2

2.91657

3.40144

0.9132

3.75

0.02551

2.55185

14.254711

3

S3

2.32648

2.68

0.9407

2.85

0.02212

2.21268

13.1906767

4

S4

3.16869

3.4635

0.888

3.89

0.03233

3.23365

8.51375733

5

S5

2.93055

3.24514

0.9174

3.53

0.02545

2.54533

9.69420259

6

S6

2.73120

2.94791

0.9086

3.24

0.03100

3.10050

7.35129068

7

S7

2.40370

2.54942

0.9311

2.73

0.02702

2.70256

5.71575695

8

S8

2.37668

2.55907

0.9132

2.79

0.03391

3.391854

7.12711242

9

S9

2.41874

2.47677

0.88

2.81

0.04845

4.84501

2.34297109

10

S10

3.52050

3.86419

4.6

0.84

0.16086

16.08695

8.61676746

Void content volume (%)

0.8

0.26

0.22

0.32

0.25

0.31

0.7

0.34

0.48

1.6

Figure 5. Synthesized TMC density in different composition of samples.

Therefore, Bulk Density of TMC achieved lower density 2.5% B4C, 7.5% SiC, 12.5% ZrO2, 77.5% Ti composition have developed minimum density of value 2.33g/cm3 and totally 24% up to 47% reduce density according to the law of mixture. Additionally, averagely reduce 38% density of synthesized TMC weight of the final product. Then, Actual Density of TMC achieved lower density 7.5% B4C, 7.5% SiC, 2.5% ZrO2, 82.5%Ti composition have developed minimum density of value 2.47g/cm3 and totally 16% up to 44% reduce the density according to the law of mixture. Additionally, averagely reduce 32% density of synthesized TMC weight of the final product.                              

Therefore, Theoretical Density of TMC achieved lower density 5% B4C,5% SiC,5% ZrO2,and 85% Ti composition have developed minimum density of value 0.7g/cm3 and totally 79% up to 85% reduce the density according to the law of mixture. Additionally, averagely reduce 72% density of synthesized TMC weight of the final product. Then, the relative density was calculated by the ratio of the actual density to the theoretical/calculated density. Then the high relative density of the sintered specimens indicates the strong interface bonding between the constituent particles with negligible porosity or cavities synthesized TMC of composite 5% B4C,5% SiC,5% ZrO2,85% Ti rating 5.17gm/cm3 achieved.

             Figure 6. comparison of actual density and density predictable by ule of mixture

As a result, the density weight percentage of the synthesized TMC was significantly reduced, and the weight of the material was also significantly reduced, making the developed material light. The main reason for facilitating weight reduction is all reinforcement low density engineering materials. Due to this reason, the result, TMC's actual density was reduced. According to the law of mixture, 7.5% B4C, 7.5% SiC, 2.5% ZrO2, and 82.5%Ti have developed a minimum density of 2.47g/cm3 and a total density reduction of 16 percent to 44 percent. Additionally, the average density of synthesized TMC weight of the final product is reduced by 32%.

3.5. Porosity measurement

The Archimedes concept was rummage-sale to compute to what extent of porosity of TMCs sintered at different temperatures. Sintered dry weight/weight in air(Wd) was measured using a precision balance for each sintered Tmc sample. After that, the fabricated samples were immersed in water and boiled for two hours before being soaked for another 24 hours. Suspension weight in water (Ww) of TMC samples was determined. The fabricated sample was immediately soaking the weight when it was rescued from the water and patted dry with tissue paper to remove excess water (Ws). The porosity was measured in accordance with Archimedes and determined by calculating using equation:[63]

                                     Eq (4)

Where:  = porosity (%), Ws = mass of sample after soaking in distilled water for 24 hours (g), Wd = mass of sintered dried sample (g), Ww = mass of sample hanging or suspension in water (g). According to [1], the porosity from theoretical density Eq. (4) is being used to evaluate its actual density of each material; hence, Eq. (4) is applied to compute the porosity of each material and the accurate porosity is calculated. Variation of Porosity calculated by Archimedes principles shown in Figure 12,                                  Eq (5), Where P represents the porosity occurring in the material, ρa represents its actual density and ρt represents its theoretical density.

  The void content volume was then computed using Agarwal and Broutman's equation [65, 66]. The content void of a composite can have a significant impact on some of its mechanical properties. Higher void contents are typically associated with lower fatigue resistance, increased vulnerability to moisture absorption and weathering, and enhanced difference or dispersed in high strength performance   properties.   The ability to estimate the quality of composites requires knowledge of void content. A good composite should have less than 1% voids, whereas a bad composite can have up to 5% void content. As a result, the void content or volume of our composite is less than one, indicating that our synthesized TMCs specimen is a good composite.                 

Figure 7. Porosity analysis of synthesized TMC

Therefore, Porosity calculated by Archimedes principles, with the minimum porosity 7.5%, B4C, 7.5%, SiC, 2.5%, ZrO2, 82.5%, Ti composition 2.34% and the maximum porosity specimen 5% B4C,5% SiC,5% ZrO2,85% Ti19.5% achieved. For porosity calculation, the minimum porosity computed using theoretical density method rather than calculating porosity by Archimedes principle. The void content of synthesized TMC nine sample have been below one, and this is indicated that the synthesized TMC are good consolidated engineering materials for the application of automotive and aerospace engineering. The high relative density of the sintered specimens indicates the strong interface bonding between the constituent particles with negligible porosity or cavities.

Comment 9

“Although the work is quite extensive, in fact it can be said that the authors performed XRD (no description !!!), SEM (for selected samples)”

Response

The paragraph has been corrected as follows:

The surface topography of the synthesized specimen was investigated using a SEM scanning to show that the SEM microstructures of the synthesized TMMC and base-Ti6Al4V specimens had a coarse lamellar + microstructure with phase separation created during sintering at high temperature and subsequent slow cooling rate, and Figure 2 shows the XRD Analysis of the synthesized sample [46]. 

Figure 5 was changed to depict a blow, figure 2.

Figure 2. SEM microstructures of machined synthesized TMMC with samples (1) S1, (2) S2, (3) S3, (4) S4, (5) S5, (6) S6,(7) S7, (10) S10

Microstructure Analysis

It is observed that sample S3 (2.5%B4C,7.5%SiC,12.5%ZrO2,77.5%Ti) by increasing the concentrations of ZrO2, decreasing B4C and SiC particles decreased the porosity and densified the surface of the Ti-based metal matrix. In most places on the surface of Ti-based metal matrix sample S3, the ZrO2 particle agglomeration is seen. Because increasing the ZrO2 particle caused agglomeration. Muharrem Pul et al..... [81] investigate that the addition of ZrO2 particles in the metal matrix composite initiated agglomeration.The microstructure of S3 shows that there is no porous structure between the 77.5 %Ti and 2.5%B4C, 7.5%SiC and 12.5%ZrO2 reinforcing particles and the bonding of the phases is very good. We observed from samples S6 and S7 are the same concentrations of ZrO2 as S3, however, due to the increasing of the concentrations of B4C and SiC, S6 and S7 are more porous microstructure. Harish et al..... [82] report that the porosity of the metal matrix composite materials in microstructure increases with the increase of the particles of reinforcing. Therefore, with the same concentration of ZrO2 and different concentrations of B4C and SiC, samples S3, S6, and S7 are different surface morphology.

Comment 2 and 3

“2. XRD results are practically not discussed, the description should be completed. The figures do not show which diffractogram corresponds to which sample.

  1. More experimental data should be provided, e.g., with XRD (exposure time, measuring step, surface preparation)”

Response

Figure 6 has been replaced by Figures 3A and 3B, which are discussed further XRD analysis below.

3.2. XRD Analysis

The elemental phases present in the manufactured samples were analyzed using XRD in accordance with the XRD working principle: Bragg's law[138]. The XRD was performed on a fully computerized powder x-ray diffractometer (XRD7000 X-RAY DIFFRACTOMETER, SHIMADZU Corporation (Japan)) at 40 KV and 30 mA. The XRD spectrum were generated at a 2 degree angle ranging from 10 to 85 degrees with a 0.02 degree step size. Continuous scanning at 3 degrees per minute for 0.40 seconds. Miller indices (hkl) are used to identify various planes of atoms, and observed diffraction peaks can be related to the planes of atoms to aid in atomic structure and microstructure analysis. When analyzing XRD data, we look for trends that correspond to crystal structure directionality by analyzing the miller indices of diffraction peaks. The crystal structure determines the position and intensity of peaks in a diffraction pattern. The fabricated samples were subjected to XRD analysis to determine whether any intermetallic compounds were formed during the sintering process[59]. When the diffractometer is linked to the X'pert data collector software, d' values are displayed directly on the diffraction pattern. These d' values were then used to identify different phases using ASTM X-ray diffraction data cards. To confirm the presence of minor precipitate phases detected by the diffraction pattern, the d' values for different phases were obtained using JCPDS cards included with the software and manually compared with the diffraction pattern of all samples[60].

Figure 3, A and B replaced and changed to the plow figure

a)

b)

Figure 3. The XRD graph of a) mild before sintering b) after sintering

Figure 3 represents the XRD graph of the titanium-based metal matrix composite powders with milling before compaction and sintering were performed. The graph shows that there is a dominance of the titanium matrix peaks, which ascribes that in the milling process there was an undesirable interfacial chemical reaction between the hybrid reinforcements and the matrix with less peak are detected with angles of 2θ=27.60 and 54.480 corresponding to (110) and (211) respectively and the rutile TiO2 (JCPDScard number: 021-1276 ) developed. As shown in Figure 3, a and b, titanium with a hexagonal closed packed crystal structure with a = b = c = 1.587 and ∝= β = γ ≠ 90  with an experimental density of 4.6 g/cm3 can be detected in the titanium metal matrix samples, regardless of whether it is before or after sintering. However, in the composite shown in Figure 3. b, the peaks corresponding to distinct phases are recognized as Ti, B4C, SiC, and ZrO2. The presence of Ti, B4C, SiC, ZrO2, and rutile (TiO2) in the titanium metal matrix is highly correlated with the presence of Ti, B4C, SiC, and ZrO2 in the titanium metal matrix. The peak of rutile (TiO2) (JCPDS card number: 021-1276) was shown for all samples at 2θ =27.440, 41.60, 44.080, 79.90 and 84.340 corresponding to the (110), (002), (210), (212) and(400) crystallographic plane respectively. The peak of SiC was shown at angles 2θ =41.280, 62.780, and 76.60   corresponding to the (200), (110), and (103) crystallographic plane (JCPDS card number: 049-1623). The peaks of B4C at angles 2θ = 69.840 correspond to the (220) crystallographic planes. The peaks ZrO2  are also shown at angles of 2θ= 27.50 and 64.10 corresponding to (-111) and (211) crystallographic planes. It is observed that after sintering of the metal matrix of all components of the composite detected by x-ray.

Round 2

Reviewer 1 Report

The scale in Figure 2 is illegible. Please enlarge it.

Dear Authors, I appreciate the corrections made. There are a few more minor remarks for improvement

Please correct the line spacing and the indentation of the text throughout the article.

LInia 41-123 has a different indentation and line spacing than line125-156.

Wrong width of text and distance between lines165-198 and 200-215, 221-286, 288-299.

Reviewer 2 Report

The authors corrected the manuscript taking into account the main comments of the reviewer.

Reviewer 3 Report

The authors have responded to my comments, mostly in a satisfactory manner. I still believe that the manuscript could be improved in terms of novelty but in its current state it is legible. After rebuilding a manuscript, it is easier to see its strengths. The authors have devoted a lot of work to improving the article. In particular, I am glad that XRD descriptions have been included and the figures have been changed. I believe the article may be published in Materials.